



# A multi-purpose, multi-rotor drone system for long range and high-altitude volcanic gas plume measurements

Bo Galle[1], Santiago Arellano[1*], Nicole Bobrowski[2,3], Vladimir Conde[1], Tobias P. Fischer[4], Gustav Gerdes[5], Alexandra Gutmann[6], Thorsten Hoffmann[6], Ima Itikarai[7], Tomas Krejci[8], Emma J. Liu[9,10], Kila Mulina[7], Scott Nowicki[4,11], Tom Richardson[12], Julian Rüdiger[6], Kieran Wood[12], Jiazhi Xu[1]

[1]Department of Earth, Space and Environment, Chalmers University of Technology, SE 41296, Gothenburg, Sweden
[2]Institute for Environmental Physics, University of Heidelberg, D-69120 Heidelberg, Germany
[3]Max-Planck Institute for Chemistry, 55128, Mainz, Germany
[4]Department of Earth and Planetary Sciences, University of New Mexico, 87131, Albuquerque, NM, United States
[5]GerdesSolutions AB, 128 41, Stockholm, Sweden
[6]Department of Chemistry, Johannes Gutenberg-University, 55099, Mainz, Germany
[7]Rabaul Volcano Observatory, P.O. Box 386, Rabaul, Papua New Guinea
[8]HAB Electronic AB, 34140, Ljungby, Sweden
[9]Department of Earth Sciences, University of Cambridge, CB2 3EQ, Cambridge, United Kingdom
[10]Department of Earth Sciences, WC1E 6BS, University College London, London, United Kingdom
[11]Quantum Spatial, Inc. Albuquerque, NM, United States
[12]Department of Aerospace Engineering, University of Bristol, BS8 1TR, Bristol, United Kingdom

*Correspondence to*: Bo Galle (bo.galle@chalmers.se)

**Abstract.** A multi-copter drone has been adapted for studies of volcanic gas plumes. This adaptation includes improved capacity for high altitude and long range, real-time $SO_2$ concentration monitoring, long range manual control, remotely-activated bag sampling, and plume speed measurement capability. The drone is capable of acting as a stable platform for various instrument configurations including: MultiGAS instruments for in-situ measurements of $SO_2$, $H_2S$, $CO_2$ and $H_2O$ concentrations in the gas plume, a MobileDOAS instrument for spectroscopic measurement of total $SO_2$ emission rate, remotely-controlled gas sampling in bags and sampling with gas denuders for posterior analysis on the ground of isotopic composition and halogens.

The platform we present has been field-tested during three campaigns in Papua New Guinea: in 2016 at Tavurvur, Bagana and Ulawun volcanoes, in 2018 at Tavurvur and Langila volcanoes and in 2019 at Tavurvur and Manam volcanoes; as well as in Mt. Etna in Italy in 2017.

This paper describes the drone platform and the multiple payloads, the various measurement strategies, an algorithm to correct for different time-responses of MultiGAS sensors. Specifically, we emphasise the need for an adaptive flight path, together with live data transmission of a plume tracer (such as $SO_2$ concentration) to the ground station, to ensure optimal plume interception when operating beyond visual line of sight. We present results from a comprehensive plume characterization obtained during a field deployment at Manam volcano in May 2019. The Papua New Guinea region, and particularly Manam volcano, has not been extensively studied for volcanic gases due to its remote location, inaccessible summit region and high level of volcanic activity. We demonstrate that the combination of a multi-rotor with modular payloads is a versatile solution to obtain the flux and composition of volcanic plumes, even for the case of a highly active volcano with a high-altitude plume such as Manam. Drone-based measurements offer a valuable solution to volcano research and monitoring



applications, and provide an alternative and complementary method to ground-based and direct sampling of volcanic gases.

# 1 Introduction

## 1.1 The use of drones for studies of volcanic plumes

The use of drones for volcanic plume studies was pioneered by Faivre-Pierret et al., (1980), who employed a fixed-wing drone equipped with in-situ sensors to measure the composition of the volcanic plume of Mt. Etna, and together with correlation spectrometry (COSPEC, Stoiber et al., 1983) derived fluxes of $H_2O$, $SO_2$, HCl and HF. This was an unmanned research aircraft that demanded high expertise and complex field operations. More recently, McGonigle et al. (2008) employed an unmanned helicopter equipped alternatively with a Multi-component Gas Analysis System (MultiGAS, Aiuppa et al, 2005; Shinohara, 2005) and a portable Differential Optical Absorption Spectrometer (MobileDOAS, Galle et al., 2003) system to measure the flux of $CO_2$ and $SO_2$ from the crater rim of Vulcano Island, Italy. This was a proof-of-concept study that demonstrated the utility of a commercial system to acquire proximal measurements, although the vehicle still required a level of piloting expertise. The first use of a multi-rotor platform capable of reaching a high-elevation plume was reported by Mori (2016), who performed measurements with a multi-rotor in the eruptive plume of Ontake volcano and measured $SO_2$, $CO_2$, $H_2S$, $H_2O$, HCl concentrations employing a combination of MultiGAS and MobileDOAS instruments. The authors reported various problems such as the need to properly shield electrochemical sensors used in MultiGAS units from electromagnetic interference derived from the drone motors or radio telemetry. Similar work was conducted by Rüdiger et al. (2018), who measured the plumes of Masaya, Stromboli and Turrialba volcanoes and studied the aging of halogenic species using drone-mounted denuder samplers. De Moor et al. (2019) and Stix et al. (2018) used a multi-rotor to perform Multi-GAS and MobileDOAS measurements and to collect plume samples for posterior carbon isotopic speciation analyses. They also measured plume speed by letting the drone drift freely with the winds at plume level. Mandon et al. (2019), also used a multi-rotor and a sampling unit to collect high-temperature filter pack samples of volcanic gases at White Island volcano, which were characterized geochemically focusing on the composition of trace metal aerosols. At Villarrica, comparison of the plume chemistry measured directly above the lava lake (using a drone-mounted gas analyser) and downwind on the crater rim (using a traditional ground-based instrument) simultaneously showed how volcanic plumes can dilute and homogenise over short distances of <150 m, especially in turbulent crater environments (Liu et al., 2019). Furthermore, Schellenberg et al. (2019) described in-plume ash collection at long-range using fixed-wing vehicles at Fuego volcano (Guatemala). This review is intended to highlight representative studies using drones as platforms for volcanic plume measurements; more comprehensive reviews, including the broader applications of drones in volcanology, can be found in Villa et al. (2016) or James et al. (2020).

The abovementioned studies show different aspects of the capabilities of drones to reach volcanic plumes and vents and perform measurements with various levels of complexity. Our study combines multiple aspects of these independent developments to show that the same unit can be used to achieve the abovementioned goals in a single field experiment, with maximum field-operability in terms of portability and reduced number of people required for the measurement, i.e. usually only two. These features make our approach a practical solution to expand the use of drones for routine monitoring of volcanic plumes.





### 1.2    Manam volcano

Manam volcano, located in Papua New Guinea, is the highest volcano in the Bismarck Arc. The volcano is a 3000 m high composite volcano that rises about 1800 m above mean sea level (AMSL). The island of Manam is about 10 km in diameter. Manam has erupted about 40 times since the early 1600s. The current phase of eruptive activity began in June 2014 and continues to date of writing, characterized by sporadic VEI4 eruptions superimposed on persistent passive degassing and minor explosive activity (Global Volcanism Program, Venzke, 2013). In 2004, an eruption devastated large sectors of the island and displaced thousands of people to the mainland. Manam is currently ranked as one of the top ten $SO_2$ and $CO_2$ emitters in the world and degassed about 1 Mt $CO_2$ $a^{-1}$ during 2005-2015 based on petrological proxies to estimate the $CO_2$ flux (Aiuppa et al., 2019; Fischer et al., 2019).

Manam is an archetypical case that represents the challenges to obtain detailed information on volcanic plumes for a large proportion of volcanoes in the world. Indeed, our present knowledge of the composition and flux of gas emissions from volcanoes is limited due to the same reasons described above: remote location, explosive activity or inaccessible vents and plumes. Although satellite-based remote sensing instruments have a large potential to overcome these limitations, this approach is in general only valid for $SO_2$ because its atmospheric background concentration is low compared to the volcanic signal. For most other species, in-situ measurements are the only option, when feasible. From a total of nearly 1500 Holocene volcanoes listed in the Smithsonian Institution's Global Volcanism Program database (https://volcano.si.edu/list_volcano_holocene.cfm), about 50% have a summit altitude above 2 km AMSL. Nearly 100 of Holocene volcanoes had a flux of $SO_2$ above the Ozone Monitoring Instrument (OMI) 1-year-average threshold of about 40 t $d^{-1}$ (Carn et al., 2017) during 2005-2016, of which ~60% have a summit at 2 km AMSL or higher. Therefore, a drone-based platform for gas sampling or real-time measurement of volcanic species in the plume needs to be suitable to reach plumes at high elevation where air density is low, and robust enough to sustain harsh measurement conditions such as acidity, corrosion and turbulence, encountered in dense regions of the plume. Thus, Manam volcano is a suitable volcano for demonstrating challenging use of a drone in volcano gas monitoring, and for this reason it was chosen as the target of a field campaign (ABOVE, Liu et al., 2020) to characterize this strong, yet difficult to access volcanic plume.

## 2    Methods

### 2.1    The drone platform

We designed our system to fulfill the following demands:

- capability to measure in-situ concentrations in excess of 1 ppm above ambient for the major components of the plume: $H_2O$, $CO_2$, $SO_2$ and $H_2S$ in real-time.

- capability to measure the flux of all major volcanic species in the plume; $SO_2$ with direct measurement and the other gases after obtaining their ratio against $SO_2$ and combination with $SO_2$ flux.

- capability to collect physical bag samples of the volcanic plume, for posterior isotopic analysis (of carbon or other species) on the ground;



- capability to reach altitudes higher than 2 km above take-off altitude, and ranges of the order of 5 km;

- relatively low cost, low expertise threshold to operate, and high field robustness and portability.

With these demands in mind, we developed the following concept: a multi-rotor drone-platform with modular payloads for different types of measurement. We chose a hexacopter in Y-shape configuration, model "Micro" developed by Sky-Eye Innovations in Sweden, which we dubbed "Munin". Among the advantages of a multi-rotor configuration we include 1) the possibility to hover, which is required for optimally positioning of the drone in the plume, for collecting samples in a bag from a confined region and for measuring during a time long enough to guarantee a good signal; 2) the possibility to perform measurements in the vertical direction, required to, for example, determine the concentration profile in the plume; 3) high maneuverability to adapt to changing wind conditions and to chase high volcanic gas concentration regions of the plume; 4) high portability due to small size, low-weight and foldable parts; and 5) simplicity for operation, usually not requiring expert piloting capabilities and no more than two people. The main disadvantage of a battery-driven multi-rotor in comparison with combustion-powered (i.e. liquid fuel) platforms is limited time-of-flight, which translates into reduced time of measurement in the plume, especially at high altitudes and distances. A main disadvantage with the combustion powered platforms related to gas volcanic measurements is risk of contamination, especially under hovering conditions.

### 2.1.1 Drone feasibility studies and lessons learned

The development of our drone platform was initially motivated by the goal to measure the $CO_2$ emission from Bagana volcano, identified as one of the strongest emitters of $SO_2$ (Carn et al., 2017), in 2016. The high level of activity of this remote volcano made it impossible for people to reach its active vents for sampling, and to reach a plume at summit level required at least 1600 m of vertical climb above ground. We tested an earlier version of our drone for high-altitude flights, in order to assess the maximum altitude achievable with a payload of about 1 kg. These tests were performed at the ESRANGE Space Center, which is operated by the Swedish Space Corporation and located near to Kiruna, in northern Sweden. Loaded flights were done in vertical and horizontal trajectories to measure the current consumption during climb (at 5 m s$^{-1}$), hovering, descent (at 4 m s$^{-1}$) and cruise (at 8 m s$^{-1}$). Typical currents for these flight modes were 54, 33, 22 and 30 A (1200 W, 730 W, 490 W, 660 W), respectively, during horizontal wind conditions of ~10 m s$^{-1}$. Vertical ascents up to 1800 m above ground were reached in these tests, above which radio control signal (at 2.4 GHz) was lost. Using two batteries, each consisting of 24 Li-Ion cells (3.7 V, 2.5 Ah each) connected in a 6S4P configuration, gives a total battery capacity of 20 Ah. For a conservative mean current consumption of 40 A, this battery capacity allows for a nominal flight time of 30 min. After the experience gained from this study and field studies in Bagana and Ulawun (NH Autumn 2016), Etna (NH Spring 2017) and Langila (NH Autumn 2018) volcanoes the main lessons learned were:

- When ascending and moving horizontally, it was found that energy consumption could be reduced if the rise and forward motion was balanced in an optimal way, as compared to moving only in one direction at a time. When the drone flew into clouds, energy consumption increased by about 50%. On descent, it was found that stability deteriorated when descending through clouds. Therefore, we avoided clouds as much as possible during flights. The drone's angle through the air, tilt related to the ratio between horizontal and vertical motion, also proved to be of great importance for energy consumption. Both





during ascent and descent, energy consumption could be minimized by considering and taking advantage of the wind's strength and direction.

- The volcanic plumes were typically found to move both horizontally and vertically within a short time span. In order to be able to sample the plume with in-situ methods it is crucial that the plume center, having the densiest gas concentrations, can be reached. Thus it is important to be able to receive real-
time information at ground of a relevant plume tracer ($SO_2$ concentration) and to be able to control the remote drone location from ground, and thereby adapt the flight path of the drone in response to this real-time data stream.

- Electrical interference from motors and telemetry influenced the noice level of the electrochemical sensors. Thus shielding and location of power and data cables, as well as location of antennas was
important (see hardware modifications detailed below).

- It was found that the time needed for switching between different payloads could be considerably reduced by changes in the drone frame and payload designs (balance, power connection, data access, telemetry).

- Access to the drone flight logs were found to be useful for post flight analysis of power consumption
with different flight modes, extraction of wind information and backup data of basic parameters like pressure, temperature and position.

Based on these experiences we modified the standard model of our platform in the following ways:

- Change of operative system: we adopted an open-source navigation module PixHawk V4 with its own power distribution card in order to overcome typical restrictions in altitude and distance of other
commercially available solutions, as well as to access all information of the flight-logs for posterior analysis.

- Modifications in the frame-design: these modifications include addition of a larger payload-carrying platform, the use of more robust motors with race drone ESC (Electronic Speed Control) that improves the maneuverability, longer propellers than are typically standard for a drone of this size (18 inch). We
also placed the batteries below, instead of on top, of the main frame to gain further stability. The drone has a triple Inertial Measurement Unit (IMU), dual compass and one GPS, and was provided with two 6S4P 10 Ah batteries.

- Control board: Jeti DC16 was used as a pilot controller and for transmission a Crossfire (TBS Crossfire Diversity Nano RX) was used. A tablet was connected to the Crossfire for flight planning and for
monitoring the telemetry.

- Increased telemetry range: we replaced the common 2.4 GHz by a 900 MHz radio link and used a high-gain directional antenna for ground-control. Figure 1 shows a photo of the drone and its main modular payloads. Technical specifications of our drone are provided in Table 1.

- Electrical interference: We used a shielded metal box for the electrochemical sensors, and tested the
optimal location of power and data cables and antennas to minimize electrical interference on these detectors.



With these modifications we have been able to reach heights of 2000 m above take off position (equivalent to absolute altitudes of up to 3700 m AMSL) and a range of nearly 5 km. Although we have limited the total flight time to 30 min under normal conditions (i.e. those conditions resulting in normal current consumption), with

favorable conditions and flight piloting strategies, a flight time of 35 min could be achieved within a safe margin. In the following sections, we describe each of the modular payloads and measurement strategies compatible with this drone platform.

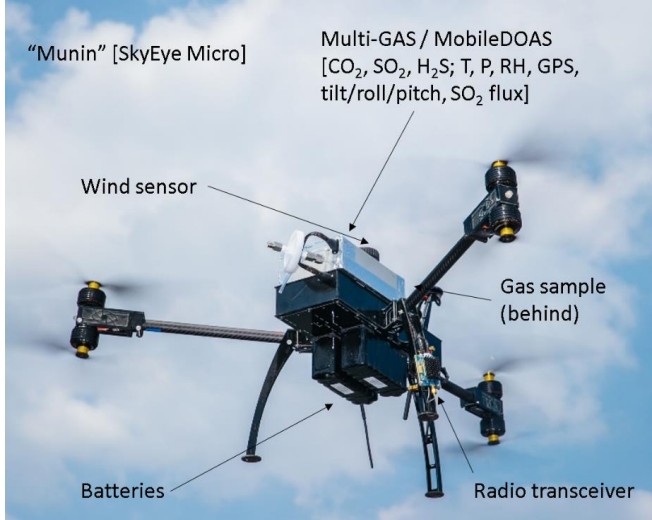

**Figure 1** Photo of the multi-rotor drone with modular payloads. The MultiGAS unit includes in-situ sensors
for gas composition, a gas-sampling unit and an anemometer. The MobileDOAS is used for remote sensing of gas flux. The modules are clamped to the drone at balanced position. The battery pack is placed below the drone chassis to lower the center of gravity of the system. Flight and sensor data are telemetered in real-time (photo courtesy of Matthew Wordell).

**Table 1. Technical specifications of the hexa-copter**

| Model | SkyEye Innovations Micro |
|---|---|
| Nickname | Munin |
| Configuration | Y6 multi-rotor (three pairs of co-axial, counter-rotating rotors in tandem) |
| Navigation system | PixHawk model V4 |
| Remote control | Jeti DC16 |
| Drone size (D×H×W) / cm | 80×20×23 |
| Frame weight, without batteries / kg | 3.0 |
| Drone weight, including batteries / kg | 4.5 – 6.0 |
| Maximum payload / kg | 2.0 |
| Maximum combined thrust/ N | 120 |



| Battery voltage / V | 22.2 (6×3.7) |
|---|---|
| Battery capacity / Ah | 20 (2×10) |
| Control range / km | 5 |
| Typical flight time (1 kg payload) / min | 30 |
| Maximum climb speed / m s$^{-1}$ | 5 |
| Maximum descent speed / m s$^{-1}$ | 4 |
| Maximum cruise speed (still wind) / m s$^{-1}$ | 10 |

### 2.2    In-situ measurements of plume speed

When determining the gas emission rate—using methods such as MobileDOAS (Galle et.al., 2003), ScanDOAS (Edmonds et al., 2003; Galle et al., 2010) or COSPEC (Stoiber et al, 1983)—information about the wind speed at plume height is critical. Since volcanic plumes are often located at several kilometers altitude and the measurements are conducted in an area relatively close to a major topographic structure, acquiring representative measurements of plume speed are challenging. The use of atmospheric models (such as operational databases provided by NOAA or ECMWF) is an alternative approach; however, these models are usually coarse in horizontal, vertical and temporal resolutions, and thus validation of these modelled wind data by local measurements is valuable; a drone can offer such in-situ validation.

We have applied two different methods for plume speed measurements using the drone: drone drift and anemometer. In the drone drift method, the drone is first positioned at the altitude of the plume and then the GPS position-locking is deactivated. The drone is thus left free to drift with the horizontal wind and, after an initial lag time of less than one minute, the drone reaches the local wind speed. The movement of the drone is logged with a separate GPS receiver, from which the local wind speed can be determined. Additionally, the actual airspeed can also be monitored in real-time through the information sent to ground control, derived from GPS data. In the second method, a small and lightweight anemometer was installed on the drone, logging the total velocity experienced by the drone (wind velocity + drone velocity). With the drone held in a fixed position close to the plume, the plume speed is thus obtained. The advantage of this method over the drone drift approach is that plume speed may be derived at the same time as other measurements in the plume are conducted. The anemometer is integrated with the MultiGAS unit, described below.

### 2.3    MobileDOAS for remote measurement of SO$_2$ emission rate

Since 2002 an instrument referred to as MobileDOAS (Galle et al, 2003; Johansson et al., 2009a) has been used increasingly to replace the previously used COSPEC instrument (Stoiber et al, 1983) for measurements of volcanic SO$_2$ gas emission rate. Both MobileDOAS and COSPEC instruments use the diffused UV solar radiation as a light source for the determination of the total column of SO$_2$ above the instrument, which is calculated using absorption spectroscopy. During a typical measurement, the instrument is moved in such a way that it passes under the gas plume in a direction close to perpendicular to the plume transport direction, while simultaneously recording spectra and GPS location. Thus, by correcting for deviations from traversing the plume perpendicularly using GPS data and using DOAS evaluation algorithms to derive SO$_2$ total columns along the





track, the total number of SO$_2$ molecules in a cross-section of the gas plume can be derived. This quantity is then multiplied by wind speed at the center of mass of the plume—i.e. plume speed—to derive the gas emission rate.

The main sources of error in these measurements are "dilution" of the absorption signal due to simultaneous collection of skylight that has been either transmitted through the plume or scattered from outside of the plume
240     (Millán, 1980) and limited knowledge of the plume speed (Galle et al., 2010).

MobileDOAS measurements from a drone platform offer several advantages compared to traditional approaches (e.g., Rüdiger et al., 2018; De Moor et al., 2018): regardless of infrastructure (roads) and topography, traverses can be made in a direction perpendicular to the plume direction; measurements can be made at an elevated altitude, thus reducing the the effects of light dilution; and plume speed can be determined reasonably accurately
245     by the methods described in section 2.3.1.

General details of the MobileDOAS hard- and software is given in Johansson et al. (2009a). Details specific to our drone-mountable version of the instrument are provided in Table A1 (Appendix A). A schematic view of the MobileDOAS instrument is shown in Fig 2.

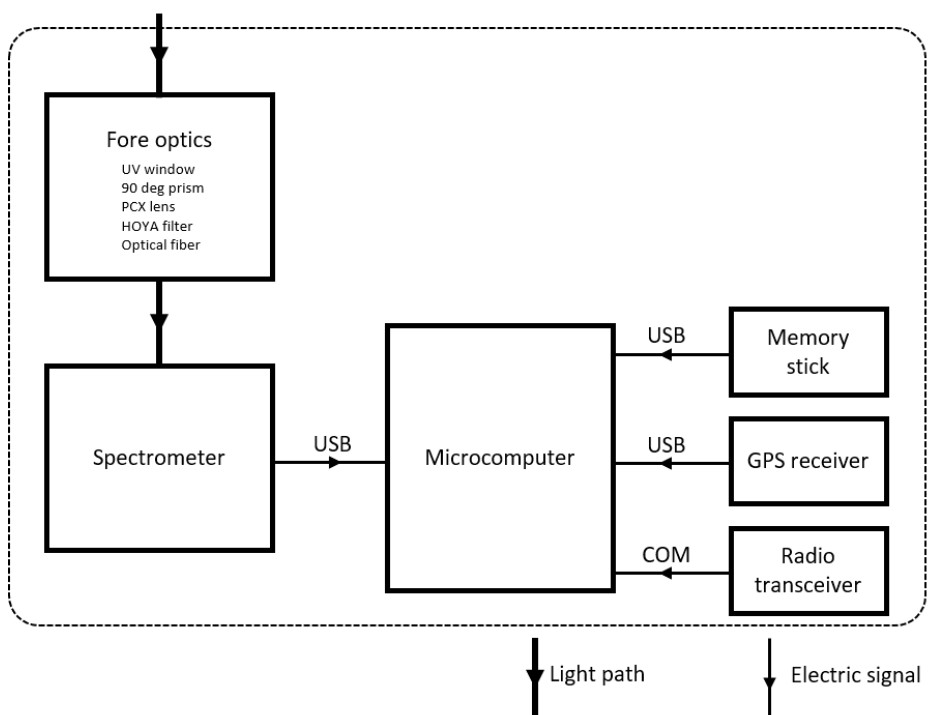

250     **Figure 2. Schematic layout of the MobileDOAS instrument**



The MobileDOAS instrument is built into a plastic case with clamps adapted to a platform on the drone in a balanced position, and it is powered from a 12/5 V power cable from the drone that was permanently active. Thus, the MobileDOAS could be installed on the drone platform, and the MobileDOAS software could be started-up or data from the MobileDOAS could be backed-up, without the main drone power turned on, thereby saving battery power. While the MobileDOAS software is active, a stream of basic information (time, position, $SO_2$ column density) is transmitted in real time via an independent radio link from the instrument. These real-time data help the pilot to guide the drone and ensure a complete the traverse of the plume. The full traverse can be visualized upon landing, by connecting an external computer to the instrument computer running MobileDOAS.

### 2.4 MultiGAS for in-situ measurement of gas composition

MobileDOAS and ScanDOAS are used to obtain $SO_2$—and under some circumstances also BrO (Lübcke et al., 2014)—emission rates from the ground using remote sensing techniques. However, to obtain the relative concentrations of other volcanic species, such as $CO_2$ and $H_2S$, direct measurements must be conducted within the plume itself; high atmospheric background concentrations or weak optical absorption of these species preclude robust detection by remote sensing methods. The most common method used for this is Multi-component Gas Analyser Systems (MultiGAS; Aiuppa et al., 2005; Shinohara, 2005). MultiGAS-type instruments generally consist of several small sensors (typically electrochemical or optical), with low power consumption, connected to a micro-computer and sometimes a data link for real time data transfer. The instrument is typically installed in a gas-exposed location (e.g. a crater rim), often proximal to the vent location. Although its installation may be labor-intensive and sometimes risky, this method is generally straightforward. However, on many volcanoes approaching the summit area would represent an enormous risk—this is the situation at Manam. In such cases, performing the in-plume measurements using a drone is an attractive possibility. An obvious requirement here is the ability of the drone to reach high altitude as well as having long endurance. This is emphasized further by the fact that many of the sensors used have slow time responses, while the gas concentrations (especially close to the vent where the signal is stronger) may vary quickly within seconds due to dilution and turbulent wind conditions. Under these conditions it is preferable to expose the sensors to the volcanic gas for as long a sampling duration as possible—ideally at least several minutes. Short-time fluctuations in concentration and plume location also imply that the ability to monitor a gas tracer (such as $SO_2$) in real-time is desirable to "chase" high-concentration sites. This is of course not guaranteed when the drone is sent in autopilot, unless an adaptive flight routine based on a plume tracer is implemented.

MultiGAS units combine information from different sensors to determine the mixing ratios of different species. In our present system, these quantities are determined according to the following processes:

For $SO_2$ and $H_2S$:

The target gas is pumped into a chamber to which the electrochemical sensors are exposed.

The electrochemical sensors are based on a differential method. The signal generated by the electrochemical effect from an electrode exposed to the gas of interest is subtracted from the signal of a reference electrode inside the system This differential signal is proportional to the gas mixing ratio. The proportionality is linear within a



certain range and depends to some degree on temperature, pressure, and the concentration of interfering species.

These proportionality and disturbance factors are determined by calibration.

For $CO_2$:

- The same gas sample is passed through a cavity illuminated by two infrared beams with wavelengths centered in and out of an absorption band of $CO_2$.

- Using the Beer-Lambert's extinction law the local concentration of $CO_2$ is determined.

295          - To get the mixing ratio, further corrections are needed for temperature, pressure and relative humidity, which should be established by calibration.

For the case of $H_2O$, the mixing ratio can be derived from measured relative humidity, pressure and temperature, following known thermodynamic laws (see Appendix B). If the measurement of such variables is done inside the sampling circuit, the $H_2O$ mixing ratio of the sample can be determined simultaneously to the other species. Our

system, however, measures these variables only inside the instrument box, so the mixing ratio is representative of ambient gas passively diffusing in the interior of the unit; $H_2O$ therefore varies differently than the other species as it is determined from outside of the closed system. The schematic layout of the MultiGAS instrument used in this study is shown in Fig 3.

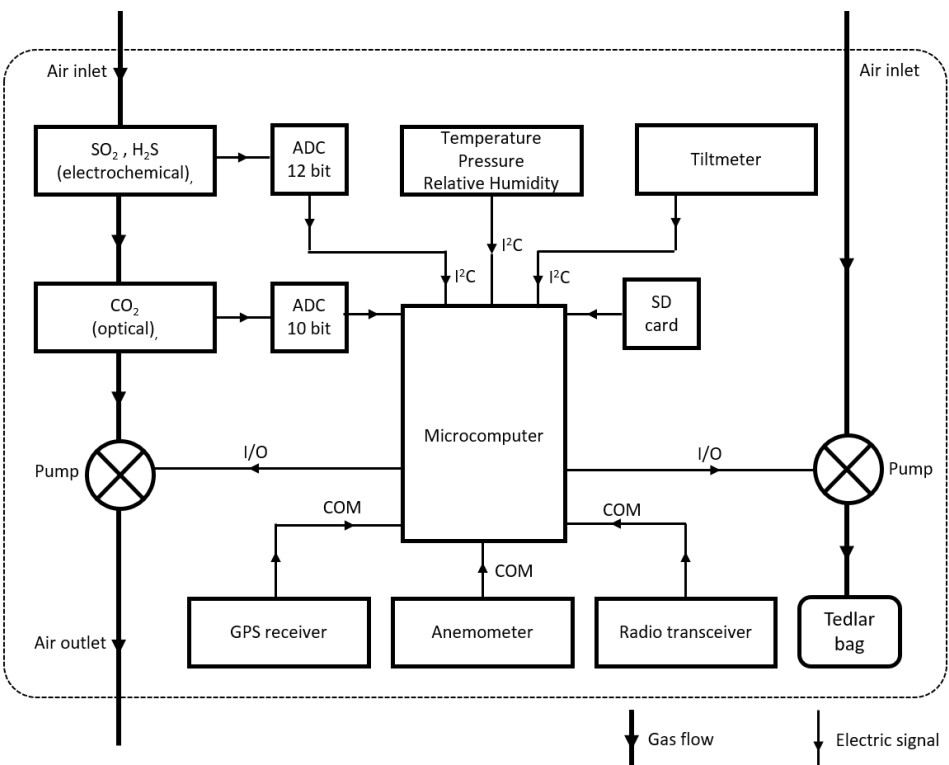

**Figure 3. Schematic layout of the MultiGAS instrument.**



Data from this unit is transmitted through an independent data link and visualized at the ground using homemade software used for tracking emissions from ships (Beecken et al., 2015; Mellqvist et al., 2018). This visualization is the basis to fine-tune the position of the drone for sampling of more concentrated regions of the plume.

To obtain the mixing ratios representative of the volcanic emission, it is necessary to correct for the contribution from the same species present in the background atmosphere. Ideally, such a measurement should be done at similar ambient conditions (i.e. pressures, temperature) to those expected inside the plume, unless the corrections for different conditions are known precisely.

When two sensors have different time response characteristics, the signals they measure will have different
amplitude and shape and be time-shifted with respect to the input signals depending on the frequency content of the input. This "distortion" of the input signals can be large when a rapidly fluctuation signal is measured for a short time, i.e. the instrument basically records only the transient signal. This could be the case with MultiGAS measurements on a flying drone in a turbulent plume. These effects must be considered to reproduce the input signal and subsequently analyse it (for example, taking the ratio of two signals such as $CO_2/SO_2$).

However, if only the ratio of the signals, and not their instantaneous amplitudes is sought, it is enough that the dynamic constants are similar. Alternatively, the true amplitudes may be obtained if the input signals have variations in time scales longer than the characteristic times of the sensors and the measurement is taken for a time longer than the response-time of the sensor. This can be achieved in two ways, either by selecting carefully the characteristics of the sensors or by exposing them to a near-constant signal. To achieve the latter, a practical
solution is to take a sample of a time-varying signal and then expose the sensors to the sampled gas for a time long enough to achieve the correct amplitudes. Our system fulfills these two criteria: the sensors have similar response characteristics and the MultiGAS incorporates a bag and pump unit that makes it possible to take samples of the plume and then expose it to the sensors for several minutes i.e. at ground. This method was however not tested in the actual field campaign, because the sampled gas was instead used for isotopic
composition analyses.

A detailed analysis of the problems encountered when combining data from sensors with different time responses is given in the following section.

### 2.4.1 Correction of time-response differences in MultiGAS sensors

When measurements are made with combined data from several sensors having different time-response, e.g.
measurements of the ratio $CO_2/SO_2$ using a MultiGAS instrument, great care must be taken.

In this measurement procedure, three characteristic times are important:

- the time of variability in gas concentration, $t_v$

- the sampling time, $t_s$

- the response time of the sensor, $t_{90}$ (meaning the time to reach 90% of the true signal for a step-change
in concentration, higher than the detection limit)

The first characteristic is determined by variability in emission, variability caused by local turbulence at the point of measurement and variability caused by relative transit of the drone with respect to the plume. The second characteristic is determined by the sampling rate of the instrument and the time required for exchange of the gas



sample inside the measurement cavity. By the third characteristic, we mean the dynamic response time of the
sensor. Because our sensors operate according to different principles, the sensor response times are usually
different; this may introduce artefacts in the mixing ratios, which would then result in wrong molar ratios for the
different species, as discussed by; Aiuppa et al., (2005); Shinohara (2005); Roberts et al., (2017) and Liu et al.,
(2019).

Drone MultiGAS measurements are normally performed by hovering in a region of high gas concentration. This
means that the relative motion of gas parcels is mostly determined by local turbulence. Farther away from active
vents, concentration heterogeneities are largely smoothed out, but the signal is very weak. Therefore, strong
signals are usually subject to high variability.

Our MultiGAS intakes a flow of 0.5 L/min ($\sim 10^{-5}$ $m^3$ $s^{-1}$), which means that for a measurement cell section of
$\sim 10$ $cm^2$, the flow-speed is in the order of 0.01 m $s^{-1}$. This results in a negligible dynamic pressure in relation to
atmospheric pressure inside the measurement cavity. The relation between the sampling and variability time is
determined by Nyquist criterion. The instrument would be able to capture accurately only signals with frequency
fluctuations lower than half the sampling frequency, (in practice a much higher sampling frequency is required).
Our MultiGAS takes a sample every second, so variations with frequencies higher than 0.5 Hz cannot be
properly captured. Such dynamic changes in plume composition are assumed to be improbable for most typical
scenarios. The $CO_2$ sensor has a cavity with dimensions of 153 mm × 30 mm × 36 mm, i.e. that the sampling
volume of 0.17 L is exchanged in a time of about 20 s. For the electrochemical sensors, the response time
depends on the transport through the membrane, dissolution in the electrolyte, reaction time and sampling. Thus,
differences in sensor geometry and measurement principle produce differences in time responses, even if the gas
flow rate remains constant (i.e. with the same pump).

The time-response depends on the dynamic properties of the sensor and the nature of the signal. For first-order
sensors, only one energy-storing and one energy-dissipating component dominate, and oscillatory behaviors are
neglected. For such sensors, the dynamical response can be modelled through the differential equation (e.g.
Pallas-Areny and Webster, 1991):

$$x(t) = a_0 y(t) + a_1 \frac{dy(t)}{dt}$$

(Equation 1)

Where x(t) is the time-varying input signal (e.g. mixing ratio of $SO_2$), y(t) is the time-varying measured signal
(e.g. voltage of the $SO_2$ electrochemical sensor), dy/dt is the first derivative of the measured signal respect to
time, and $a_0$ and $a_1$ are constants identified with the time-response ($\tau = a_1/a_0$) and static sensitivity of the sensor
($K = 1/a_0$).

The dynamical time-response factor defines a delay in the response of the measured signal in relation to the input
signal. If the input signal is a step-function, one usually relates this factor by the time required for the sensor to
achieve a certain level of the signal, for example 90%. Periodical signals will be affected by an error in
amplitude and by a shift caused by the frequency response of the sensor. The steady-state amplitude response to
a signal of angular frequency $\omega$ is given by $k/[(\omega\tau)^2+1]^{0.5}$. The shift is given by $\tan^{-1}(\omega\tau)$. An arbitrary signal





can be represented by a Fourier sum of periodical signals, and for linear systems, the response of the sensor is
obtained by superposition of the responses to the monochromatic signals.

We correct our signals based on these conditions:

- That sensors of the MultiGAS instrument can be accurately modelled as first-order systems.

- That the input signals of different sensors measured at the same time are highly correlated.

- Variability of the gas concentration occurs at a characteristic time much shorter than the exchange time
in the sensor.

- The total measurement time is much larger than the exchange/diffusion time in the sensor.

The first assumption is supported by the design and laboratory characterization of the electrochemical and optical sensors. The second assumption requires that the molar ratio of different species is constant and homogeneous within the time of measurement. Sampling a heterogeneous mixture would produce different ratios at different times, complicating both the measurement and the interpretation of the results. In volcanic emissions,
drastic changes in molar ratios within minutes are unlikely if the gases come from the same source. But if the plume mixes emissions from different vents or if large local heterogeneities affecting unequally the chemistry or condensation of different species (e.g. for plumes with heterogeneous concentration of ash), changes in gas molar ratios can occur even on short time scales. The third condition ensures that enough information is
available for finding a unique solution, and the fourth condition is required to reduce the error caused by sampling over a different exchange/diffusion times of the sensors.

Based on the first assumption, our method starts with the two measured signals (e.g., $y_{CO2}(t)$ and $y_{SO2}(t)$) and constructs from them their derivatives ($dy_{CO2}(t)/dt$ and $dy_{SO2}(t)/dt$) by simple numerical approximation, which works fine as long as the sampling time is short (i.e. as long as large variability in time scales shorter than the
sampling rate is not expected). Based on the second assumption, we expect that a simple scaling exists between the two input signals, defining a constant ratio r ($r = x_{CO2}(t)/x_{SO2}(t)$). Now we simply iteratively vary the time--response factors of the two signals and look for the combination that maximizes the cross-correlation between the reproduced inputs according to Equation 1. This method works best for strongly fluctuating signals, rich in information for the correlation analysis. Fig 4 shows an example of this method applied to field measurements of
a very dynamical signal obtained in the crater of Tavurvur volcano (Papua New Guinea) in 2016.

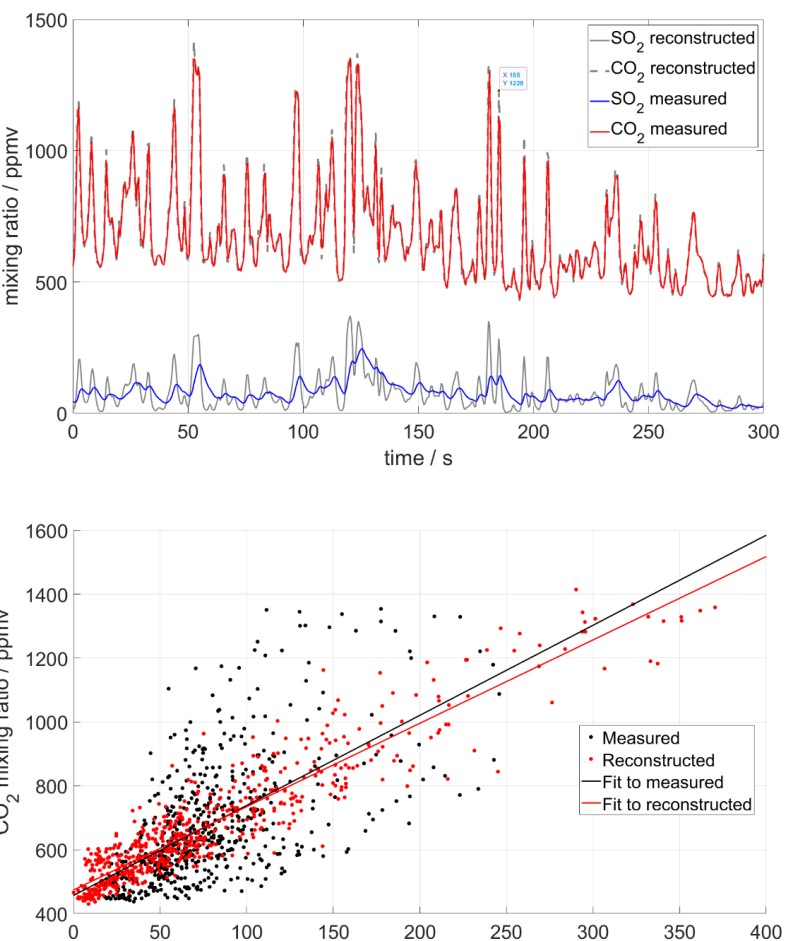

**Figure 4. (a)** Time series of $CO_2$ and $SO_2$ mixing ratios measured inside the crater of Tavurvur volcano in September 2016 using a MultiGAS unit (Arellano et al., 2016). The two signals are re-constructed iteratively until an optimal correlation is found between them. For this instrument, the $SO_2$ sensor was slower than the $CO_2$ sensor and could not capture all rapid fluctuations. **(b)** After the correction, the correlation between the two series is higher and the dispersion of the scatter plot much lower. But most importantly, the $CO_2/SO_2$ molar ratio changes from 2.8 ± 0.3 to 2.6 ± 0.08 (± of 95% CI), and the background $CO_2$ (inside the crater) changes from 457 ± 26 ppm to 473 ± 9 ppm


This method is quite general for the correction of the dynamic response of the sensors. It obviates time-consuming and frequent characterization of the time response in a laboratory and accounts for the fact that the sensors may change their dynamic characteristics when exposed to different conditions in the field, relative to the lab. Of course, calibration is still desirable to check for possible changes in offset and sensitivity of the sensor over time.



### 2.5 Bag sampling unit for gas composition and isotopic analysis

In some cases, real time measurements in the gas plume are not possible either because longer measurement time is needed to reduce the signal-to-noise ratio  or because the measurement requires an analytical technique that is more complex than can be performed in situ on a drone platform. In such cases, acquiring a sample of the plume gas using a drone carrying a sample bag and a pump may be a viable option. For successful sampling of the most concentrated region of the plume, real time data transmission of a plume tracer—such as $SO_2$—to the drone operator on the ground (in combination with manual flight control to respond accordingly) is advantageous; however, this can be technically challenging, especially for long-range flights.

Measurement of the carbon isotope composition of volcanic $CO_2$ is a key example of an application that requires in-plume sample collection. Carbon isotope analyses are performed using instrumentation such as isotope ratio mass (IRMS) spectrometers (Chiodini et al., 2011; Sharp, 2007), bench-top infrared spectroscopic analyses (Fischer and Lopez, 2016; Rizzo et al., 2014; van Geldern et al., 2014) or cavity ring down spectroscopy (Lucic et al., 2015). While the bench top spectroscopic techniques are much more portable than IRMS systems and do not require a vacuum, they still depend on a stable 110 or 220 V power source and pressurized calibration and dilution gases—faciliaties that are not always available in remote field locations. The spectroscopic techniques generally require a sample volume of about 300 ml at atmospheric pressure and temperature. This can be accomplished by directly placing the instrument in the volcanic plume (Rizzo et al., 2014; Rizzo et al., 2015) or by collecting a plume or fumarole sample in an appropriately sealed and non-reactive container (i.e. Tedlar sample bags) for subsequent analyses by spectroscopy (Fischer and Lopez, 2016). The application of drones for this purpose is advantageous over the use of helicopters due to the lower operational cost and smaller-scale logistics, as well as avoiding sample contamination by gases present in the helicopter exhaust (Fischer and Lopez, 2016).

During our fieldwork at Manam volcano, we used two payloads to sample the plume. In the first, we equipped our drone with four Tedlar sampling bags. Each bag was connected to a small rotary pump triggered by a timer. The drone operator positioned the drone in the plume, at which point the sample was collected by a timed trigger. The duration of sample collection was approximately 45 seconds at an approximate flow rate of 1 L/min. A valve system was not necessary because the pump also functioned as a valve once it stopped pumping. The second system was similar but here the pump (at 0.5 L/min) could be remotely controlled and only one Tedlar bag was used. After return of the samples to the ground, the valves on the Tedlar bags were closed and the samples analyzed by a Delta Ray infrared spectrometer. In addition to collecting samples from the plume, a clean air sample was collected upwind and at the same elevation as the plume. In principle, the analytical procedure followed that described in Fischer and Lopez (2016). Due to the remote location of Manam island and the difficulty encountered wihen obtaining calibration and $CO_2$-free air gases in-country, we developed an air purification system that utilized a bicycle pump and $CO_2$ scrubber, Sulfolime™ obtained from PP systems Inc., to produce pressurized $CO_2$-free air. This system allowed the production of essentially unlimited amounts of $CO_2$-free air with $CO_2$ contents of <0.7 ppm, as measured using the Delta Ray. The calibration gas was pure $CO_2$ obtained from a local distributor. Prior to analysis, the C isotope composition of this gas was not known, and we therefore collected a sample of this gas to analyze back in the Volatiles Laboratory at the University of New Mexico using the Delta Ray and standard calibration gases. Therefore, we were not able to determine the exact C



isotope compositions of the samples in the field but were able to adjust the pure $CO_2$ calibration gas to the concentration of $CO_2$ measured in the bag sample by setting the corresponding $CO_2$ concentration in the Delta Ray software. We then retroactively corrected all our measurements using the values obtained for the field calibration gas.

### 2.6  Plume sampling of halogens using a denuder system

Besides $H_2O$, $CO_2$ and $SO_2$, halogens are among the major constituents of volcanic emissions (Textor et al., 2004). The discovery of bromine monoxide (BrO) in volcanic plumes and the correlation of the simultaneously gained $BrO/SO_2$ ratio with volcanic activity by automatized instruments (Lübcke et al., 2014) made $BrO/SO_2$ a promising volcano monitoring tool. To utilize $BrO/SO_2$ ratios for monitoring volcanic activity an understanding of ongoing bromine chemistry in the plume is essential. For a detailed discussion on bromine chemistry in

volcanic plumes see Gutmann et al. (2018) and references therein. Besides BrO no other bromine species can be detected by remote sensing instruments. Therefore in-situ methods have been developed recently for the determination of reactive bromine (BrX) and hydrogen Bromide (HBr) (Rüdiger et al., 2020; Gutmann et al., 2020). The new methods reveal the downwind conversion of released HBr to other bromine species with time in a volcanic plume.

Bromine speciation can be measured using gas diffusion denuder systems. Gaseous molecules are derivatized in-situ by an organic coating at the inner walls when pumped through gas diffusion denuders. Analysis of bromine speciation has been carried out for two different bromine species. HBr has been determined by 5,6-epoxy-[1,10]-phenanthroline coated denuders (Gutmann et al., 2020) and BrX (such as $Br_2$, BrCl, HOBr) and reactive chlorine species (ClX) were detected by 1,3,5-trimethoxybenzene coated denuders (Rüdiger et al., 2017). Samples were

analyzed by high-performance liquid chromatography or gas chromatography coupled to mass spectrometry, at the University of Mainz, Germany after returning from the field.

For the detection of $SO_2$ a compact MultiGAS-type system called "Sunkist" (Rüdiger et al., 2018) was used, which contained an electrochemical CiTiceL 3MST/F sensor. The calibration of the electrochemical $SO_2$ sensor followed Arellano et al. (2016). The "Sunkist" also measures ambient pressure and temperature. Since the

thermometer is inside the isolated box and is affected by running instruments, only temperatures at the beginning of the flight were considered. The temperature at the starting position was 26 °C for the first flight and 32°C for the second. Assuming a vertical temperature gradient of -5 K km$^{-1}$ and an approximately flight height of 2000 m, 16°C and 22°C were estimated for calculations, respectively.

Denuders were connected by PTFE tubes to a micro pump providing 200 mL min$^{-1}$ for each denuder. Denuder

sampling does not provide time-resolved samples and instead yields an average concentration for the whole exposure interval (i.e. an average per flight). Since results obtained with environmental denuders predict that these bromine species usually only appear within the volcanic plume, we calculate their plume concentrations based on the known duration of exposure in the plume as detected by the $SO_2$ sensor. Fig 5 shows a photo of the drone, with denuders and Sunkist, mounted on the drone and ready for take-off.

A blank correction for denuder results was performed by subtracting analysis results from coated denuders that travelled alongside samples but did not sample any air.





Background environmental blanks were sampled at the starting position of flights. For subtraction of background environmental blanks, the atmospheric concentration of samples (ppb) was calculated for the total flight time. After blank subtraction, the atmospheric concentration of the bromine species was calculated for the duration in

plume (estimated based on $SO_2$ signals).

Limits of detection (LOD) and limits of quantification (LOQ) were calculated via 3- and 10-times deviation of the coated denuder blanks (n=3) respectively. LOD and LOQ are dependent on sampling time and the processing method in the laboratory and are therefore, calculated for each sample separately.

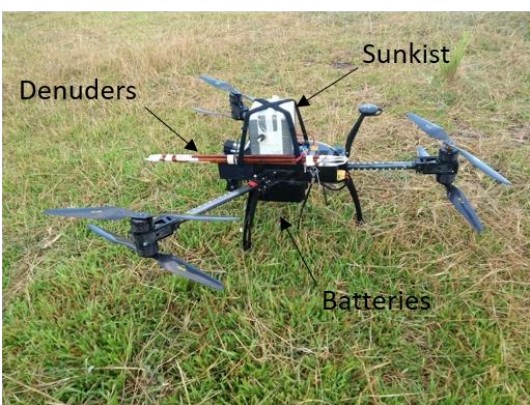

**Figure 5: Drone-based denuder sampling setup and $SO_2$ sensor (Sunkist)**

## 3     Results

### 3.1     Manam field-campaign conditions

A field campaign was conducted at Manam volcano in Papua New Guinea during 19 - 27 May 2019. This field

campaign utilized all abovementioned measurement and sampling techniques and was the culmination of several previous campaigns during which parts of the system had been extensively tested and modified in repsonse to lessons learned (see section 2.1.1). The meteorological conditions during the campaign at Manam were characterized by low wind speeds and varying wind direction. Volcanic gas was emitted from two different locations close to the summit; the Main crater and the Southern crater (Liu et al., 2020). The Southern crater was

the most active, with incandescent lava visible within the crater and emanation of a persistent strong gas plume with high buoyancy (2–3 km above the crater rim). In contrast, Main crater showed more fumarolic degassing, generating a weaker plume with reduced buoyancy (see Fig 6). Due to the low wind speed and varying wind direction there was during some periods a buildup of high ambient $SO_2$ concentrations covering a large part of the island, giving rise to challenging measurement conditions.



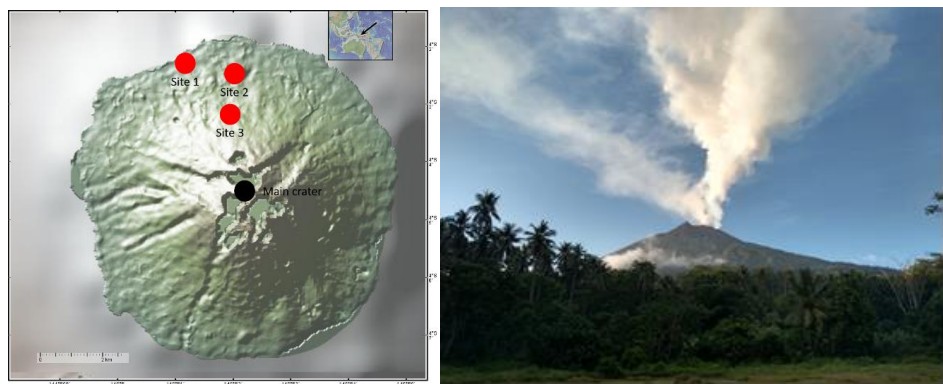


**Figure 6. (a) Map showing the location of Manam volcano (based on GeoMap App, Ryan et al., 2009), with the main sites of launch of our drone (red) and the location of the crater emissions (black). (b) Photo of the plumes of Manam taken on the 27 May 2019 from Site 1**

**3.2    Plume speed measurements**

Figure 7 shows wind velocity measured with an anemometer on-board the drone, made in connection with a MultiGAS measurement on 22 May 2019. The high wind speeds measured before reaching the maximum altitude are due to a combination of wind velocity and horizontal drone velocity when approaching the plume region. After reaching the plume region (indicated by increased $SO_2$ concentration around 2000 m altitude), the

drone is kept at a fixed position, and after a short time a stable plume speed of $3.0 \pm 0.5$ m s$^{-1}$ is obtained.

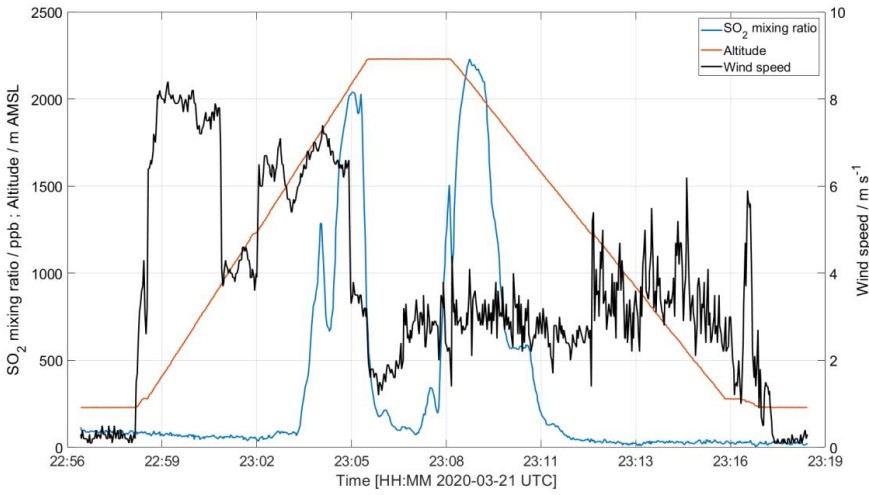

**Figure 7. Wind speed measurement made 22 May 2019 using an onboard anemometer. The effective measurement of wind speed at a certain level is obtained when the drone is placed in a hovering position for several seconds, usually in combination with composition measurements or gas sampling**






Figure 8 shows a plume speed measurement made using the drone drift method. After reaching the plume altitude the drone GPS is deactivated remotely and after a time-lag (caused by the inertia of the drone) a stable drift speed of 6.2 m s$^{-1}$ is obtained.

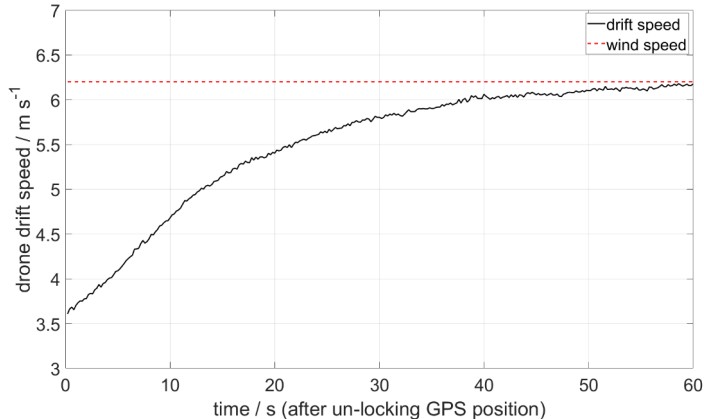

**Figure 8. Example of a plume speed measurement made by the drone drift method on 27 May 2019. The sequence starts at the moment when the GPS position is unlocked, and the drone is left free to be drifted by the wind at an altitude of 2170 m AMSL. It took 1 minute to stabilize at about 6.2 m s$^{-1}$**

As can be seen in Fig 9 the wind speed at Manam during the field campaign was variable and relatively low, 1 –
4 m s$^{-1}$, except on the final day (27/05/2019). This variability results in larger overall emission rate error estimates, as a change of a few m s$^{-1}$ in wind speed generates a large uncertainty in the emission. However, the data from the ECMWF regional model is in relatively good agreement with our drone measurements.

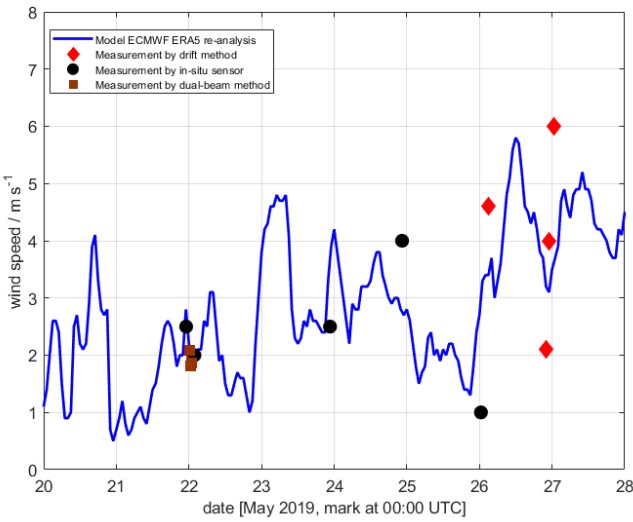





**Figure 9. Wind data at 1800 m altitude at Manam volcano showing model data from the ECMWF ERA5 model and drone data using the drone drift and in-situ anemometer methods, as well as results from a ground-based dual-beam remote-sensing method (Johansson et al, 2009b). The drone measurements were taken at different altitudes (between 1500 and 2500 m AMSL), where a $SO_2$ signal was detected by the MultiGAS sensor**


### 3.3    $SO_2$ emission rate


Figure 10 shows an example of a MobileDOAS traverse made at 1000 m altitude at Manam volcano on 27 May 2019. An $SO_2$ emission rate of $5200 \pm 660/180$ t d$^{-1}$ was obtained for this traverse using a wind speed at plume height of 6 m s$^{-1}$ measured with the drone drift method about 30 minutes after completing the traverse.

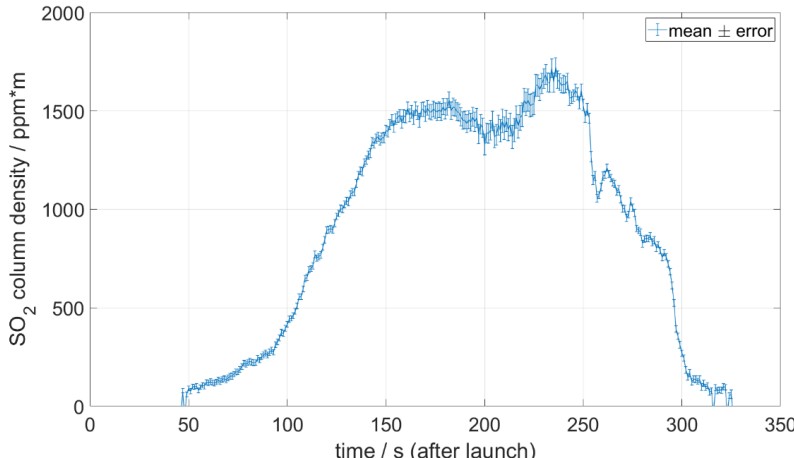


**Figure 10. Example of MobileDOAS traverse made at Manam volcano on 27 May 2019 at an altitude of 1000 m AMSL. The emission rate of $SO_2$ was 5200 ± 660/180 t d$^{-1}$, using the plume speed shown in** Fel! Hittar inte referenskälla.**. Information about the position of the drone below the plume is telemetered in real time**


This drone-based emission rate measurement can be compared to ScanDOAS measurements made from the ground on the same day, yielding an average over the day of $4512 \pm 2230$ t d$^{-1}$, using wind speed from the ECMWF ERA5 model (Liu et al, 2020). One possible reason for this higher value in the drone measurement is that the relatively high elevation minimizes the atmospheric scattering "dilution" in this measurement. Over the the full period of the field campaign, the representative $SO_2$ flux was estimated at $5150 \pm [733/336]$ t d$^{-1}$ (high and low 1-σ bound on uncertainty) by synthesising a large number of measurements (and their respective uncertainties) from ground-based, drone-mounted and satellite-based approaches (Liu et al., 2020).


### 3.4    Molar ratios

Figure 11 shows the results from a flight simultaneously carrying a MultiGAS instrument and a plume sampling device on the drone. To obtain the $CO_2/SO_2$ ratio it was first necessary to compensate for (a) the pressure and temperature effects on the raw concentration data, (b) the dynamical responses of the sensors, and (c) the



atmospheric background concentration of $CO_2$ (see details in Appendix B), where the background is identified as the concentration of $CO_2$ taken at the same altitude outside of the plume (where the plume tracer $SO_2 = 0$). The resulting $CO_2/SO_2$ ratios are shown in Fig 12. Using a linear regression, a $CO_2/SO_2$ ratio of $0.9 \pm 0.2$ is obtained for the Manam plume on 26 May 2019. For the entire campaign in 2019, the average plume molar ratio was $1.07 \pm 0.06$ based on multiple measurements acquired on different days; these additional measurements were

obtained using a different (but essentially similar in operation) MultiGAS unit onboard a fixed-wing UAV. The full dataset and volcanological discussion thereof is presented in Liu et al, (2020).

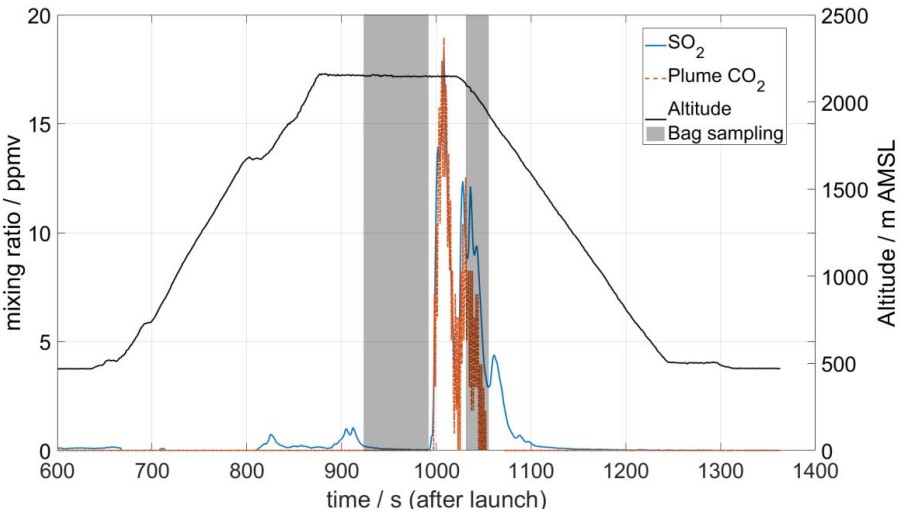

**Figure 11. MultiGAS measurements of $CO_2$ and $SO_2$ mixing ratios at 500 m above the Southern Crater of Manam volcano on 26 May 2019. $SO_2$ was detected in small concentrations above the Main Crater (region**

**around 800-900 s after launch) but peaked above the high emission column of the Southern Crater. $CO_2$ concentrations above background values are only detected over this crater. The shaded area corresponds to periods of bag-sampling activated remotely and used for isotopic analysis at Baliau Village by Delta Ray Infrared Mass Spectrometer.**

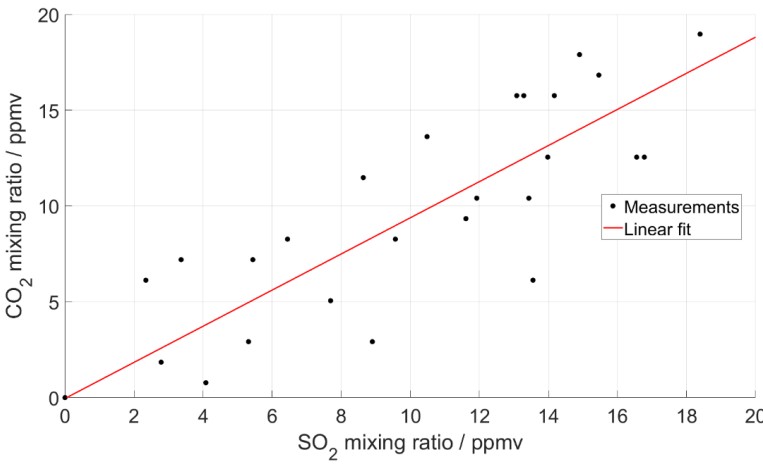



**Figure 12. A regression plot of data from** Fel! Hittar inte referenskälla. **yielding a CO₂/SO₂ ratio of 0.9 ± 0.2.**
**Only corresponding pairs where the SO₂ mixing ratio was higher than 2 ppm and the altitude was stable**
**(>2140 m AMSL) were used for the regression**

### 3.5    Carbon-isotopic composition

Figure 11 shows data from a flight conducted on 26 May 2019 combining a MultiGAS instrument with a bag
sampling unit. A first bag sample was collected after reaching the anticipated plume location, although no
significant $SO_2$ concentration was seen in the real time data at ground. After some maneuvering of the drone the
plume was found, and a second bag sample was collected.

Most samples obtained from Manam had concentrations similar to the clean air background with $CO_2$

concentrations ranging from 408 to 415 ppm $CO_2$. These samples had variable and unreliable $\delta^{13}C$ values
ranging from -4.2 to -8.6 ‰, reflecting the large error on these low concentration samples. Only five samples
were obtained with concentrations above 420 ppm, ranging from 421 to 494 ppm $CO_2$; these data and the
volcanological interpretation thereof are discussed in Liu et al, (2020). Extrapolation of these data to 100% $CO_2$
following the methodology of (Fischer and Lopez, 2016; Rizzo et al., 2014) yields a value of $\delta^{13}C$ = -3.7 ‰,

close to the MOR mantle range of -5 ± 1 ‰ of (Marty and Zimmermann, 1999) and -6.5 ± 2 ‰ of (Sano and
Marty, 1995). We note, however, that the regression is unconstrained at high $CO_2$ due to the low $CO_2$
concentrations of the plume samples; therefore, statistically the potential extrapolated range could be as large as -
3.7 ± 9.5 ‰ in the 95% confidence interval, i.e. a very large uncertainty (Liu et al., 2020). Values above +1‰
are highly unlikely given that the global compilation of arc gases has a $\delta^{13}C$ range of +2 ‰ to -19 ‰ (Fischer

and Chiodini, 2015; Mason et al., 2017; Oppenheimer et al., 2014). The lowest possible value within that range
of uncertainty would be -13.2 ‰. We can, therefore, rule out a significant contribution from organic carbon,
which would have $\delta^{13}C$ values of around –30 ‰ (Sano and Marty, 1995). Our data is generally consistent with
$\delta^{13}C$ values of fumarole and hot spring gases obtained during the few prior studies in the PNG region (-2.7 ± 0.1
‰, Sano and Williams, 1996). Additional data from more concentrated samples (with higher $CO_2$ concentrations

ideally >1000 ppm) collected closer to the vent would improve the accuracy of the Manam $\delta^{13}C$ values.
Nevertheless, most importantly these results demonstrate a valuable proof-of-concept that highlights the
potential of drone-based sampling of volcanic emissions for future geochemical studies.

### 3.6    Halogen concentrations

On 27 May 2019, we achieved two successful flights with the deunder system where detection of $SO_2$ indicated

that we had entered the plume. In both flights we detected $SO_2$ in two positions, each at different altitudes,
consistent with the observation of two source regions for summit emissions (Fig 13). The maximum $SO_2$
concentrations were 1.74 and 1.87 ppm in flight 1 and 1.22 and 4.62 ppm in flight 2 for peak 1 and 2
respectively. Detected $SO_2$ concentrations were averaged for each time we entered the plume. $SO_2$ concentration
for each flight were averaged proportionally to the sampling duration in plume (Table 2).





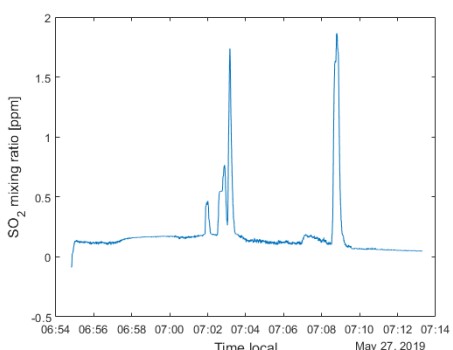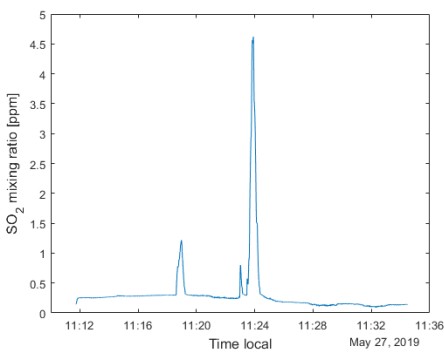


**Figure 13: SO₂ data detected by "Sunkist" in two flights on 27ᵗʰ of May 2019**

For both denuder types, an environmental blank was sampled at the starting position on the ground. For HBr the
denuder results were within the deviation of unsampled but coated denuders. For reactive bromine, the
environmental blank values were the highest values of the whole campaign (BrX: 0.2 ppb for 3 L sampling

volume). In addition, the amount of reactive chlorine was higher than that of reactive bromine in this sample
only (ClX: 0.4 ppb for 3 L sampling volume). Therefore, we infer that this sample was either contaminated or
the location in the grass close to the sea does not represent the atmospheric conditions encountered during the
flight.. Alternatively, we subtracted an environmental blank sampled 10 days earlier in a pre-campaign at

Tavurvur volcano, at an altitude of 104 m upwind on the flank of the volcano and above barren volcanic rock (17
May 2019, BrX: 0.002 ppb, ClX: 0.000 ppb). The result of the environmental blank of HBr was within the
deviation of coated denuders not used for sample collection.

Average concentrations of all analyzed bromine and chlorine species based on the duration of exposure in the
plume and their calculated LOD and LOQ are given in Table 2. Results were just below the LOD for both

bromine species in flight 1. Bromine results for flight 2 were below the LOQ, but above LOD. ClX was below
LOD in both samples. The ratio of the bromine species to SO₂ varies between the two flights from 30 x10⁴ to
44 x10⁴ for HBr/SO₂ and 3 x10⁴ to 5 x10⁴ for BrX/SO₂ from flight 1 to 2, respectively, while the ratio between
the two bromine species BrX/HBr stays stable (flight 1: 0.10, flight 2: 0.12).

**Table 2. Bromine speciation based on the length of stay in the plume detected by the SO₂ sensor**

| Date | 27.05.2019 | Flight 1 | Flight 2 |
|---|---|---|---|
| Start Time | [Local] | 06:54 | 11:11 |
| Average SO₂/both peaks | [ppm] | 0.6 | 1.0 |
| HBr/SO₂ | × 10⁻⁴ | 30 | 44 |
| Δ HBr/SO₂ | × 10⁻⁴ | 13 | 9 |
| BrX/SO₂ | × 10⁻⁴ | 3 | 5 |
| Δ BrX/SO₂ | × 10⁻⁴ | 3 | 3 |
| BrX/HBr | × 100 | 10 | 12 |





| Δ BrX/HBr | × 100 | 12 | 6 |
|---|---|---|---|
| HBr | [ppb] | 1.7 | 4.6 |
| Δ HBr | [ppb] | 0.7 | 0.9 |
| LOD | [ppb] | 4.3 | 3.5 |
| LOQ | [ppb] | 14.3 | 11.7 |
| BrX | [ppb] | 0.2 | 0.5 |
| Δ BrX | [ppb] | 0.2 | 0.3 |
| LOD | [ppb] | 0.2 | 0.2 |
| LOQ | [ppb] | 0.6 | 0.6 |
| ClX | [ppb] | 0.1 | 0.3 |
| Δ ClX | [ppb] | 0.2 | 0.2 |
| LOD | [ppb] | 0.5 | 0.5 |
| LOQ | [ppb] | 1.8 | 1.8 |


Although measurement results were below the LOQ, we were able to detect HBr and BrX in the plume of Manam volcano with our two flights. The observed BrX/HBr ratio of 0.10-0.12 are relatively low compared to previous detected ratios of approximately 0.30 within the first minute in Masaya's plume (Rüdiger et al., 2020). Observed bromine evolution in Manam plume seems to follow the trend for high Br/S-ratios suggested by
Roberts et al. (2014).

## 4     Conclusions

We present a hexa-copter UAV specially configured for long-range high altitude volcanic gas plume measurements, together with a suite of of drone-mountable payloads specialised for various gas measurements. We describe how the drone and its payloads were operated during a field-campaign at Manam volcano in Papua
New Guinea, demonstrating the the utility of the proposed system for achieving long-range high altitude volcanic gas monitoring.

In response to several key lessons learned during prior field-based testing, we implemented specific customisations to the drone platform:

-     Use of stronger motors and longer propellers to increase payload and battery weight capacity, and
660          thereby improve range and height performance.

-     Change of operative system to overcome altitude and range restrictions and enable access to flight logs for posterior analysis.

-     Use of a race drone ESC (Electronic Speed Control) to improve maneuverability during flight.

-     Change in frame design to facilitate swapping of modular payloads.

The different payloads include:



- An anemometer for measurement of wind/plume speed at plume height

- A MobileDOAS instrument that, combined with plume speed data, can measure total $SO_2$ emission rate by traversing below the plume.

- A MultiGAS instrument that can measure in-situ concentrations of $SO_2$, $CO_2$ and $H_2S$ in the plume, by flying close to the crater.

- A bag sampling unit by which the gas in the plume can be sampled in Tedlar bags for subsequent analysis of carbon isotopes at the ground.

- A denuder system that can sample the plume for later analysis of different halogens in the laboratory.

During the one week field campaign at Manam volcano we used all these instrument payloads to perform a total
of 19 successful flights to the plume of Manam (1 for $SO_2$ flux, 4 for $CO_2/SO_2$ molar ratio, 9 for wind speed, 3 for C isotopic analysis and 2 for halogen composition). All these measurements could be carried out with only two people (one flying the drone and one supervising the measurements). Figure 14 shows the trajectories of all flights for which data is presented here.

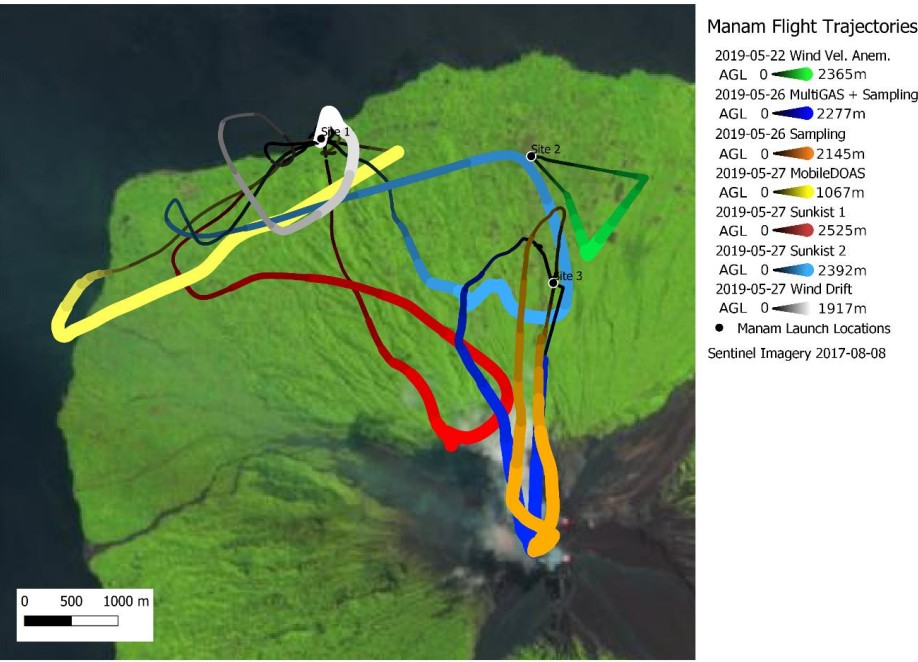

**Figure 14. Map with flight trajectories of measurements reported in this study. Wind measurement flights occurred from launch Sites 2 and 1 on May 22 and 27 respectively. Isotopic sampling and MutiGAS flights launched from Site 3 on May 26. MobileDOAS and Sunkist measurement flights launched from Site 1 on May 27. Sentinel imagery shown was selected based on minimal cloud cover and may not represent the location of vents and plumes as was present in May 2019. The**
**thickness of the trajectory markers is proportional to the altitude of the drone**



The results presented here represent a sub-set of the total dataset obtained during a collaborative international field campaign at Manam volcano, involving multiple research teams and instruments. A complete synthesis of all measured data and their uncertainties, together with detailed volcanological interpretation, is presented in Liu et al., (2020). The fluxes of $SO_2$ and $CO_2$ were found to be $5200 \pm 660/180$ and $3220 \pm 500/90$ t $d^{-1}$, respectively,
based solely on the multi-rotor measurements reported in this study. The $\delta^{13}C$ of $-3.7 \pm 9.5$ ‰ suggests that the $CO_2$ source of Manam crater gas is mantle dominated with a possible carbonate contribution and a likely insignificant contribution from subducted sedimentary organic carbon. While the range of possible values is large due to the low $CO_2$ concentrations and hence large extrapolation to pure $CO_2$, these data are generally consistent with a measured plume C/S ratio of $0.9 \pm 0.2$, indicating a predominantly mantle derived carbon
source. The bromine speciation and concentration in the plume points to a high Br/S ratio.

It was found that an adaptive flight path is essential for successful measurements, through either fully manual control at long distances or adaptable automated waypoint missions. Access to real-time data of a plume tracer, such as $SO_2$, is crucial, both when flying visual and when flying beyond visual line of sight, in order for the pilot to refine the position of the drone within the plume. Further, prior information about the approximate position of
the plume from ground-based instruments, such as ScanDOAS, is very informative during the flight planning stage to maximise the chance of successful sampling, and should be incorporated into field workflows. Finally, to facilitate long range and high altitude, under limited power capacity, the flight path in approaching and returning from the target is crucial to conserve battery; flying in clouds should be avoided, wind speed and direction should be taken into account and used to their advantage, and the angle between horizontal and vertical
motion should be optimized for low power consumption.

We have demonstrated that the combination of a multi-rotor with modular payloads is a versatile solution to obtain the flux and composition of volcanic plumes, even for the case of a highly active volcano with a high-altitude plume such as Manam. We propose that drone-based measurements offer a valuable solution to volcano research and monitoring applications at inaccessible volcanoes globally.

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

**Funding**

The main funding for this work comes from the Aerial Based Observations of Volcanic Emissions (ABOVE) project, funded by the Alfred P. Sloan Foundation through their support of the Deep Carbon Observatory (DCO) project. The drone and Chalmers payloads were supported by FORMAS, the Swedish Research Council for Sustainable Development. SA was supported by the Swedish National Space Agency (Dnr 149/18). KW was 885 supported by the NCNR EPSRC grant (EP/R02572X/1). EJL acknowledges support from a Leverhulme Early Career Fellowship. T.R was supported by the EPSRC CASCADE programme grant, (EP/R009953/1).

**Author contribution**

SA, NB, TF, AG, BG, GG, EL, KM, SN, JR, TR, and KW participated in the field measurements. GG piloted the drone. SA and BG collected and analysed Chalmers data (MultiGAS and MobileDOAS). TF and SN 890 collected and analysed UNM data (Isotope analysis). NB, AG, TH and JR collected and analysed UM data (Sunkist and denuders). GG and TK built the drone. SA, VC and JZ built and calibrated the Chalmers payloads. SN built the UNM payload. NB, AG, TH and JR built and calibrated the UM payload. EJL coordinated the field campaign with support from II, KM, TR and KW. All authors contributed to writing the article.

**Conflict of Interest**

The authors declare that they have no conflict of interest.

**Acknowledgments**



We thank the Rabaul Volcano Observatory (RVO) and the community of Baliau village on Manam Island for their assistance and hospitality. We thank the PX4 autopilot development team for their open source flight control software.

**Data Availability**

The datasets generated for this study can be provided upon request sent to the "Corresponding Author".

**5      Appendices**

**5.1      Appendix A. Technical specifications of the payloads.**

**Table A1. Specifications of payloads of the drone system**

|  | MobileDOAS | MultiGAS+Anemometer | Bag sampling unit | Denuders+Sunkist |
|---|---|---|---|---|
| **Measurement mode** | Up-looking, traverse remote sensing | In-situ sensing | In-situ sampling | In-situ sampling |
| **Main purpose** | Flux of $SO_2$ | Molar ratio of $CO_2/SO_2$ and $H_2S/SO_2$<br>Gas sampling (for post-analysis)<br>Wind velocity | Collect samples for posterior analysis of isotopic and gas composition | Gas sampling (for post-analysis of molar ratio of $HBr/SO_2$ and $BrX/SO_2$) |
| **Measured quantities** | $SO_2$ column density distribution<br>Plume speed | $SO_2$ mixing ratio<br>$H_2S$ mixing ratio<br>$CO_2$ concentration<br>Relative humidity<br>Temperature<br>Pressure<br>Air velocity<br>Br compounds<br>$\delta^{13}C$ | $\delta^{13}C$ | $SO_2$ mixing ratio<br>HBr mixing ratio<br>BrX mixing ratio<br>Pressure |
| **Size (L□W□H) / cm** | 22×15×9 | 21×14×14 | Inside MultiGAS | Sunkist 14×13×14<br>Denuders 50×3×3 + micropump |
| **Weight / kg** | 0.8 | 1 | Inside MultiGAS | Sunkist 0.5<br>Denuders 0.15 |
| **Power and voltage/ W, V** | 12, 5 V | 4, 12 V | Inside MultiGAS | Sunkist: 9V for sensor, 3.7 LiPo for Arduino<br>9V for micropump |
| **Radio link** | 400 MHz | 400 MHz | 400 MHz | NA |
| **Components** | Micro-computer (Azulle Quantum | Micro-computer (Arduino | Micro- | Micro-computer |



| | Access Mini PC Stick)<br>UV grating spectrometer (OceanOptics Flame) Telescope (25 mm diameter, FOV 11 mrad)<br>Quartz optics (window, prism, fiber)<br>GPS antenna (GlobalSat BU353S4) | Mega2560)<br>Electrochemical sensors (Alphasense $SO_2$-A, response time 20 s, Alphasense $H_2S$-A, response time 20 s)<br>Dual-band IR radiometer (SmartGas FlowEVO $CO_2$, response time 30 s)<br>Enviromental sensor (Bosch BME280, RH, T, P)<br>GPS antenna (Adafruit Ultimate)<br>Anemometer (FT205EV)<br>Pump (0.5 L min$^{-1}$)<br>Tedlar bags (1 L) | computer (Arduino)<br>Pump (1 L/min)<br>4 Tedlar bags (X L) | (Arduino with micro SD card logger)<br>Electrochemical sensors (CiTiceL 3MST/F, response time < 20 s (SO2)<br>Environmental sensor (BMP180 by Bosch Sensortec)<br>Pump (200 mL /min/Denuder)<br>Denuders (brown borosilicate glass tubes, 50 cm, id 0.7 cm) |
|---|---|---|---|---|
| **Sampling frequency / Hz** | 1 | 1 | On/Off set by timer | 2 ($SO_2$)<br>Denuders: no time resolution, 1 sample/flight |
| **Precision** | 5 ppm*m ($SO_2$)* | 2 ppb ($SO_2$)<br>1 ppm ($CO_2$)<br>2 ppb ($H_2S$)<br>3% (RH)<br>0.01 K (temperature)<br>0.2 Pa (pressure)<br>0.1 m s$^{-1}$ (wind speed)<br>1 deg (wind direction) | 0.15‰ ($\delta^{13}C$, analyzed by Delta Ray IR spectrometer) | |
| **Accuracy** | 1 ppm*m ($SO_2$)* | 15 ppb ($SO_2$)<br>1 ppm ($CO_2$)<br>5 ppb ($H_2S$)<br>3% (RH)<br>1 K (temperature)<br>120 Pa (pressure)<br>0.3 m s$^{-1}$ (wind speed)<br>4 deg (wind direction) | ± 0.5‰ ($\delta^{13}C$) | ± 1 ppm ± 1% signal (SO2)<br>Halogen species: For each sample separately (see Table 2). |
| **Range** | 0-20000 ppm*m ($SO_2$) | 0-50 ppm ($SO_2$)<br>0-1000 ppm ($CO_2$)<br>0-50 ppm ($H_2S$) | 200 ppm – 2500 ppm $CO_2$ | 0–200 ppm (SO2)<br>Halogen species: LOD and LOQ for each sample |





| | | 0-100 % (RH) | | separately (see Table 2). |
| | | 243 – 358 K (temperature) | | |
| | | 300-1100 hPa (pressure) | | |
| | | 0-75 m s$^{-1}$ (wind speed) | | |
| | | 0-360 deg (wind direction) | | |

*For S/N~500:1 assuming 50% of intensity saturation for an average of 15 spectra taken at 0.5 s exposure time.

### 5.2    Appendix B. Analysis of MultiGAS data

To obtain molar ratios of different species measured by different sensors, the general procedure followed Equation B1:

$$X_A = \alpha_A + \beta_A \times DN_A + \tau_A(T_{Am} - T_{AC}) + \pi_A(P_{Am} - P_{Ac}) \qquad \text{(Equation B1)}$$

Where:

$X_A$, molar mixing ratio of molecule A ($CO_2$, $SO_2$, $H_2S$)

$\alpha_A$, offset of sensor for molecule A, from lab calibration in ppm

$\beta_A$, sensitivity of sensor for molecule A, from lab calibration in ppm/DN

$DN_A$, digital number measured by sensor for molecule A

$\tau_A$, temperature correction factor of sensor for molecule A, from manufacturer's specifications, in K

$T_{Am,c}$, temperature measured/calibrated by thermometer in parallel to sensor for molecule A, in K

$\pi_A$, pressure correction factor of sensor for molecule A, from manufacturer's specifications, in ppm/K

$P_{Am,c}$, pressure measured/calibrated by thermometer in parallel to sensor for molecule A, in Pa

After these corrections, the time-response correction is performed by the procedure described above and the ratio determined by simple linear regression on a scatterplot, using only data with $SO_2$ mixing ratio higher than 2 ppm.

For water, the mixing ratio is obtained from the following equations (Wagner and Pruß, 2002):

$$X_{H2O} = 10^6 \times \left(\frac{P_W}{P-P_W}\right) \qquad \text{(Equation B2)}$$

$$P_W = 10^{-2} \times RH \times P_{WS} \qquad \text{(Equation B3)}$$

$$P_{WS} = P_C \times exp\left(\frac{T_C}{273.15+T}\right) \times f(\theta) \qquad \text{(Wquation B4)}$$

Where:

$X_{H2O}$, water dry molar mixing ratio

$P_W$, water vapor pressure, in Pa

$P$, measured ambient pressure, in Pa

$RH$, relative humidity, in %



$P_{WS}$, water saturation vapor pressure, in Pa

$P_C$, water critical pressure, in Pa

$T_c$, water critical temperature, in K

$T$, measured ambient temperature, in K

$f(\theta)$, a function of $T_c$ and $T$ involving six parameters

MultiGAS data is processed with Matlab.

### 5.3    Appendix C. Analysis of MobileDOAS data

To obtain the $SO_2$ flux, the integral of $SO_2$ column densities across a surface perpendicular to the plume
direction is multiplied by plume speed. Plume direction is obtained from the vector joining the source (crater)
and the peak column density in the traverse. Plume speed is obtained preferably by the drone-drift method.

To obtain the $SO_2$ column densities, each spectrum in the traverse is divided by a spectrum from a region outside
of the plume (which could be identified a-posteriori as one with minimum column density). The transmittance
spectrum is fitted to a model following the DOAS method (Platt and Stutz, 2008). The model includes absorption
cross-sections of $SO_2$ (Vandaele et al., 1994) and $O_3$ (Voigt et al., 2001), a synthetic Ring-effect pseudo-absorber
(from Chance and Kurucz, 2010), and a polynomial of $5^{th}$ order to account for broad-band extinction due mainly
to scattering. The fitting is performed between 310 and 325 nm and shift is allowed for a spectrum with high $SO_2$
signal and then fixed for the rest of spectra in the same traverse.

Data is processed with the MobileDOAS software (Johansson, Yang and Norgaard, 2019).