# Peer review of "A multi-purpose, multi-rotor drone system for long range and high-altitude volcanic gas plume measurements"

_Atmospheric Measurement Techniques, 2020_

## Referee Comment (RC1) · Peter Kelly (Referee) · 22 Jan 2021

Summary (please see the attached pdf supplement for full review with specific comments)

This article describes a small, electric-powered multi-rotor drone and several payloads that were used for volcanic gas sensing and sampling at volcanoes in Papua New Guinea between 2016-2019. The authors focus on technical descriptions of the payloads (DOAS, multi-GAS, a denuder system, and gas-bag collection system) and modifications that were made to the drone platform to improve its endurance. This contribution appears to serve as a technical companion paper to (Liu et al., 2020), who discuss

the volcanologic significance of the obtained gas composition and emission rate results from the 2019 campaign.

Some of the payloads used in the experiments have been described previously (e.g. the DOAS system, denuder system, and 'Sunkist' instrument; Rüdiger et al., 2018), but the manuscript does include descriptions of a new multi-GAS unit developed by Chalmers U. that includes the innovative integration of a mini anemometer to obtain windspeeds, as well as a plume sampling unit for collecting bagged samples for posterior carbon-isotope analysis. To me, the most novel aspect of the manuscript is the presentation and analysis of the two methods for determining plume speed; most of the other instruments and techniques have been in use for some time. Accurately determining plume speeds is critical for determining volcanic gas emission rates, and the instrument and methods comparison shown here are helpful for addressing this important issue.

The technical emphasis of the manuscript is appropriate for Atmospheric Measurement Techniques and the operational 'lessons learned' will be valuable and of interest to the volcanic gas community. The manuscript is generally well-written and structured but there are some items that need to be addressed prior to publication. Broadly, my main concerns (documented below) are that the manuscript is too vague in places, and that supporting data are incomplete, contain mistakes, or are not available in an open repository. The scientific value of the collected gas measurements are hardly discussed (perhaps a little more effort could be made here, or would it overlap too much with Liu et al.?), therefore I feel that the technical contribution must be significant and substantive to warrant publication. These issues compromise the study's impact and value in its present form but should not be too difficult to remedy. The article will be appropriate for publication in AMT after these issues and the comments below are resolved. I hope that these comments are helpful.

Peter Kelly, USGS

Please also note the supplement to this comment:
https://amt.copernicus.org/preprints/amt-2020-452/amt-2020-452-RC1-
supplement.pdf

––––––––––––––––––––––––––––––

[Figure]

**Supplement:**

Review of "A multi-purpose, multi-rotor drone system for long range and high-altitude volcanic gas plume measurements"

Submitted to Atmospheric Measurement Techniques (AMT)

Bo Galle[1], Santiago Arellano[1], Nicole Bobrowski[2,3], Vladimir Conde[1], Tobias P. Fischer[4], Gustav Gerdes[5], Alexandra Gutmann[6], Thorsten Hoffmann[6], Ima Itikarai[7], Tomas Krejci[8], Emma J. Liu[9,10], Kila Mulina[7], Scott Nowicki[4,11], Tom Richardson[12], Julian Rüdiger[6], Kieran Wood[12], and Jiazhi Xu[1]

Lead author affiliation:
[1]Department of Earth, Space and Environment, Chalmers University of Technology, SE 41296, Gothenburg, Sweden

Review by Peter Kelly, USGS, Jan 20, 2021

**Summary**

This article describes a small, electric-powered multi-rotor drone and several payloads that were used for volcanic gas sensing and sampling at volcanoes in Papua New Guinea between 2016-2019. The authors focus on technical descriptions of the payloads (DOAS, multi-GAS, a denuder system, and gas-bag collection system) and modifications that were made to the drone platform to improve its endurance. This contribution appears to serve as a technical companion paper to (Liu et al., 2020), who discuss the volcanologic significance of the obtained gas composition and emission rate results from the 2019 campaign.

Some of the payloads used in the experiments have been described previously (e.g. the DOAS system, denuder system, and 'Sunkist' instrument; Rüdiger et al., 2018), but the manuscript does include descriptions of a new multi-GAS unit developed by Chalmers U. that includes the innovative integration of a mini anemometer to obtain windspeeds, as well as a plume sampling unit for collecting bagged samples for posterior carbon-isotope analysis. To me, the most novel aspect of the manuscript is the presentation and analysis of the two methods for determining plume speed; most of the other instruments and techniques have been in use for some time. Accurately determining plume speeds is critical for determining volcanic gas emission rates, and the instrument and methods comparison shown here are helpful for addressing this important issue.

The technical emphasis of the manuscript is appropriate for Atmospheric Measurement Techniques and the operational 'lessons learned' will be valuable and of interest to the volcanic gas community. The manuscript is generally well-written and structured but there are some items that need to be addressed prior to publication. Broadly, my main concerns (documented below) are that the manuscript is too vague in places, and that supporting data are incomplete, contain mistakes, or are not available in an open repository. The scientific value of the collected gas measurements are hardly discussed (perhaps a little more effort could be made here, or would it overlap too much with Liu et al.?), therefore I feel that the technical contribution must be significant and substantive to warrant publication. These issues compromise the study's impact and value in its present form but should not be too difficult to remedy. The article will be appropriate for publication in AMT after these issues and the comments below are resolved. I hope that these comments are helpful.

**Data availability**

At present the manuscript does not adhere to the data standards for AMT. Line 901: 'The datasets generated for this study can be provided upon request sent to the "Corresponding Author".' The data from this study needs be made open and accessible, in accord with current community and journal standards.

AMT/EGU Data policy:
https://www.atmospheric-measurement-techniques.net/policies/data_policy.html

Unfortunately, the data that are available appear to be incomplete and contain mistakes. For example, I tried to further examine the multiGAS data presented in the study but encountered significant difficulties in attempting to do so, as described below.

MultiGAS data from the experiment are said to be available from Liu et al., 2020 (l. 580-581):

https://advances.sciencemag.org/content/suppl/2020/10/26/6.44.eabb9103.DC1

Data that I was able to find in the supplement to Liu et al. (2020) includes data from three flights on May 22 and May 23, 2019 but not from other dates (for example, data from 2016 are plotted in Figure 4 and data from May 26 are plotted in Figure 11 and are not in the Liu et al. supplement). The data available in the supplement apparently include data from two multigas instruments: one from U. Palermo and the Chalmers instrument described herein (although the supplement does not readily indicate which data came from which instrument, or I missed the explanation somewhere – my apologies if I simply missed it!). The supplementary data also do not include absolute timestamps (only sequential integer seconds) so it's not possible to precisely connect these data to the results listed in the Liu et al. study by date/time, and while they do include lat/long, the altitudes are missing which makes plotting and understanding the flight paths hard.

Since Liu et al. (2020) emphasize the data from the U. Palermo instrument, my best guess is that the first two tabs include data from that instrument and my hunch is that the third dataset (Raw data 23-05-19 B) came from the Chalmers instrument, but of this I am not positive. My hunch that the presented data come from different instruments is supported by the observation that the data formats are different (e.g. the first two data tabs have lat/long listed as decimal degrees and the third has lat/long as UTM), but I really don't know for certain.

If my hunch is correct, then what I take to be Chalmers multiGAS data from May 23 appear to be very poor, and only show two $SO_2$ peaks near the end of the file (one of which is partially truncated; shown below) and $CO_2$ is poorly correlated with $SO_2$. The poor correlation between the $CO_2$ and $SO_2$ would appear to violate assumption 2 of the analysis routine (l. 383). Elsewhere the $CO_2$ data show large apparent jumps of ~15 ppm; is this some kind of interference? Finally I note that no units are given for any of the measurements, and the air temperature (Tair) appears to be listed in A/D counts(?) rather than sensible units (the 'Tair' values range from 5369 to 6805). I could not find data from the Sunkist unit anywhere.

[Figure]

*Figure showing CO₂ and SO₂ data from Liu et al., 2020 supplement 'abb9103_Table_S2', perhaps showing data from the multiGAS unit described in the present work(?). Note that the second SO₂ peak is truncated and that CO₂ and SO₂ are poorly correlated.*

It's possible that these are the wrong data but based on the available information it is very difficult to be sure.  This example makes it clear that the data from this experiment should be:

- better organized
- documented
- quality-assured
- checked for completeness
- made available in an open repository

At present, it is not possible to easily or unambiguously find or identify data from the described systems, which makes independent evaluation of their performances impossible.  When preparing these data for release, please make sure that all necessary parameters (e.g. altitude) are included.

**Specific Comments**

1. The range achieved in the present work (~ 5 km) is good for a multirotor system but is not especially noteworthy and has been achieved previously with commercial multi-rotor drone systems, as summarized in James et al., 2020, Table 2 (copied below) and the references therein (quote from James et al.): "UAS equipped with miniaturised gas sensing instrumentation (see Section 2.2.3; Figures 2A, 15) are now bridging the gap between direct measurements and remote sensing observations, enabling repeatable, proximal measurements from ranges >5 km [Pieri et al., 2013a; Shinohara, 2013; Diaz et al., 2015; Mori et al., 2016; Xi et al., 2016; Di Stefano et al., 2018; Rudiger et al., 2018; Stix et al., 2018b; Kazahaya et al., 2019; Liu et al., 2019; Schellenberg et al., 2019; Syahbana et al., 2019]."

**Table 2: Example vehicle characteristics.**

| System | Payload weight (total weight) | Flight endurance (min.) | Approx. range (km) | Reference |
|---|---|---|---|---|
| *Single rotor* | | | | |
| Thunder Tiger Raptor 90 | 3 kg (8.2 kg) | 12 | 16 | McGonigle et al. [2008] |
| RMAX-G1 | ~10 kg (94 kg) | ~90 | ~5 | Kaneko et al. [2011]; Koyama et al. [2013]; Ohminato et al. [2017]; Kazahaya et al. [2019] |
| *Multi-rotor* | | | | |
| LAB645 | 4 kg (12 kg) | ~40 | ~4 | Terada et al. [2018] |
| Vulcan UAV X8 | 800 g (~10.5 kg) | 12–18 | 2–4 | Liu et al. [2019]; Pering et al. [2019] |
| αUAV | 2.5 kg | ~35 | ~4 | Mori et al. [2016] |
| Phantom 4 Pro | (1.4 kg) | <25 | | Manufacturer specifications www.dji.com/phantom-4-pro |
| *Fixed-wing* | | | | |
| VectorWing 100 | 1 kg (3.6 kg) | 30–45 | 10–15 | Pieri et al. [2013a] |
| Skywalker Titan | 1.0 kg (8.5 kg) | 30–45 | 5–10 | |
| Skywalker X8 | 0.2 kg (4.2 kg) | 30–45 | 5–10 | Schellenberg et al. [2019] |

Some of the UAS listed in James et al. are gas-powered, but, for example, Syahbana et al., 2019 used a small electric-powered fixed-wing drone that launched from 11 km distance and gained > 3000 m altitude to monitor Agung Volcano during unrest and eruption. Endurance to launch from >10 km is significant because the vast majority of deaths from PDCs at explosive arc volcanoes occur within ≤10 km distance from the vent (Brown et al., 2017). The introduction should reflect the fact that small, electric, multi-rotor drones cannot match the range possible with equivalent fixed-wing units, but they do have significant advantages in terms of maneuverability, ease of take-offs and landings, ease of integrating payloads, etc., as discussed in the manuscript.

The notion of what constitutes 'long range' and 'high-altitude' is subjective, but given that the drone's performance more or less matches the capabilities of other similar systems it may be appropriate to consider changing the title to the present work to "A multi-purpose, multi-rotor drone system for volcanic gas plume measurements".

2. Figure 1 of the manuscript is copied verbatim from Liu et al. (2020) supplement Figure S3b (shown below) and without correct attribution. Although the team for the two papers are much the same, 'recycling' figures in this way is poor practice and - at minimum - any previously published images and/or figures need to be cited properly.

[Figure]

Figure S3: Annotated diagrams of the Unoccupied Aerial Systems (UAS). (a) Fixed-wing 'Titan' aircraft. Image credit: K. Wood; (b) Multi-rotor 'Munin' aircraft in Y6 configuration. Image Credit: M. Wordell. See Material and Methods for more details.

*Figure S3 from Liu et al. (2020) supplement. Liu et al's Figure S3b is copied and presented as Figure 1 in the present work without correct attribution.*

l.149-156: These seem like very important operational observations. However, critical details are left out or are too vague. "The drone's angle…also proved to be of great important for energy consumption." Can this be made more specific and actionable?  What is an optimal flight vector?  Roughly how much endurance might the flight optimization gain?  Why did power consumption go up so much when encountering clouds?

l. 166-168: "It was found that the time needed for switching between different payloads could be considerably reduced by changes in the drone frame and payload designs (balance, power connection, data access, telemetry)."  The present description is too vague. What 'changes' were made?  How were instruments mounted?  How was balance checked?

l. 169-171: "Access to the drone flight logs were found to be useful…"  I agree that they would be very useful for the reasons stated.  Please make the logs available as part of the data release.

l.284-296: On my first reading it was unclear if the $CO_2$, $SO_2$, and $H_2S$ sensors were self-made or commercially-available units.  Only upon reading Appendix A it became clear that the sensors were not self-made.  Please amend the section to include specifics of the sensors so that readers don't need to consult the Appendix to find this. Also, the descriptions of the NDIR and electrochemical sensors can be shortened since their measurement principles are well known and described in the manufacturers' documentation.

Figure 3: the box diagram shows the anemometer located inside the instrument enclosure.

3. Measurement of in-plume $H_2O$ mixing ratio

l.299-301 If I understand correctly, the RH sensor included in the multigas unit measures RH inside the instrument box, not in the sample gas stream. If true, the instrument therefore does not achieve its claimed capability of measuring in-plume $H_2O$ (l. 26).  The authors seem to implicitly acknowledge this detail in the conclusions (l. 669) where they list $CO_2$, $SO_2$, and $H_2S$ as measured gases instead of $H_2O$, $CO_2$, $SO_2$, $H_2S$ as claimed earlier and in the abstract.  It's

confusing to me why the authors go on to discuss how to calculate in-plume $H_2O$ mixing ratios when their system doesn't actually have that capability: l. 299-301 "Our system, however, measures these variables only inside the instrument box, so the mixing ratio is representative of ambient gas passively diffusing in the interior of the unit.." I'm very sorry if I've misunderstood something but I've read through lines 297-303 several times now, and the description and Figure 3 indicate that T, P, and RH measure conditions inside the sampling box, not in the sample stream or in ambient air outside the instrument enclosure. While it's perhaps acceptable to use the P record as 'ambient' or 'near-ambient' P, the T and RH will be useful as diagnostics, but not as plume or ambient measurements. As presented, this is very confusing. I recommend clarifying which in-plume measurements the system supports, and which measurements are for diagnostics. Since the RH appears to be intended for diagnostics, then there is really no need to discuss conversion to in-plume $H_2O$ mixing ratio and that text can be deleted.

Furthermore, the desired resolution of the water measurement is purported to be 1 ppmv (l. 110) - an ambitious goal, to be sure - but the precision of the RH sensor is stated as 3% RH. The point is moot since it appears that the instrument was not designed to measure plume $H_2O$, but for the sake of argument let's say the total P = 1000 hPa, the $P_{H2O}$ is 20 hPa, and saturated vapor pressure is 25 hPa. In this case the RH would be approximately 80%. Here, ±3% RH precision would translate to about ±0.75 hPa or about 750 ppmv (it's unclear if the given precision is 1σ or a range, etc., please clarify here and throughout). This example suggests that in practical terms it would be impossible to achieve 1 ppmv $H_2O$ resolution with the specified sensors. In addition to random and systematic errors on the RH measurement, I would expect some error in the relationship used to convert RH to mixing ratio, random and systematic errors in the needed P+T measurements, etc.

Please carefully edit the manuscript so that the measurements made are characterized accurately, and that realistic analytical values are given as design goals, and that the methods used to characterize the accuracy and precision of the various sensors are described and/or listed in the measurement specifications. For example, the denuder section (2.6) gives a clear statement on how LOD and LOQ were calculated (l. 501-503).

l.326-327: Please include the time constants of the sensors here. Are the values listed in the appendix $t_{90}$? Please clarify.

L.327-329: This is an interesting idea, but I do have concerns about its viability. Would a ~1L tedlar bag provide enough gas to get a good 'plateau value' from the sensors? How long do the sensors take to plateau during calibrations? Also, at low concentrations I would expect some sorption of S-containing species that could impact the results.

l.337: what is "the time of variability in gas concentration"? The meaning is not clear in the explanation.

l.360-361: 20 seconds to exchange the volume of the $CO_2$ optical cell seems like an awfully long time. Can the pump rate be increased to shorten this time? It would be very useful to see what a step function looks like during calibration of the $CO_2$, $SO_2$, and $H_2S$ sensors.

4. Multigas data processing technique

l.365-366: "…only one energy storing and one energy dissipating component…" I think what's being referred to here is a resistive-capacitive circuit (RC), which are classically described as first order systems. Perhaps consider recasting this section using more standard terminology.

Equation 1: this looks like an interesting approach; amplifying the signal to better approximate the input versus instead of lowpassing a 'fast' sensor to match a slower sensor. I couldn't get a copy of the reference within the timeframe of this review. Are there other more easily-accessed references that explain this theoretical approach? Is $\tau$ the first order time constant here? Most of the electrochemical sensors I use have $\tau$ values between about 2 and 6, so would $a_1$ normally be 2 to 6 and $a_0 = 1$?

l.379: I appreciate that the equations for the frequency-dependent amplitude and phase lag are given, but they could use a little more explanation for readers and a reference. At minimum, the angular velocity ($\omega$) should be defined and it should be made clear that these relationships describe the phase shift in radians (I suppose that $\tan^{-1}(\omega\tau)*180/pi$ could be given if degrees are preferred).

l.382: Please provide evidence that shows the responses of your sensors (e.g. many studies show calibration peaks).

l.383: I am concerned that the assumption that the input signals are highly correlated is dangerous; this may work for measurements with high signal to noise and for homogenous plumes, but heterogenous plumes exist and sometimes ambient background variations can be significant (e.g. Kelly et al., 2013). This is a significant weakness of the outlined approach.

l.384-386: It's not clear to me what these assumptions mean. Is there another way to clarify?

l.397-405: A worked example is needed here, perhaps as part of the data release. It's not enough to say that the evaluation happens in matlab. If the intent is for others to try to use this technique, then an example dataset would be most useful. When iterating the time response factors, is only one parameter varied ($a_1$) and $a_0$ set to 1?

Figure 4: I'm surprised the $CO_2$ shows so much variation with the 20 second time necessary to exchange the gas in the optical cell. Also the response time in the Appendix is listed as 20 seconds for the $SO_2$ and 30 seconds for the $CO_2$, yet the $CO_2$ appears to be much 'faster' than the $SO_2$ sensor. How is it possible that the $CO_2$ sensor has such a long exchange time, slower response, and yet shows much sharper measurements than the $SO_2$? The $CO_2$ also shows much smaller corrections than the $SO_2$ sensor. Can this be explained and clarified or corrected? What values of $a_0$ and $a_1$ delivered the optimal fit?

The corrected signal in 4b still shows considerable scatter (as do the data in Figure 11), which suggests that perhaps the model isn't working so well. It would be useful to compare the results of this method to the approach described by Roberts et al. (2014).

l.416-420: I disagree that characterizing sensor responses is overly time-consuming. Characterizing such responses offers many advantages for tracking sensor health and provides a basis for simple, reproducible, and automated data processing routines that require no assumptions of plume homogeneity like the presented method. What evidence exists that labderived time responses differ from field performances?  This section feels like an overreach and the claims should be substantiated or revised.

l.462: What was the value of the calibration gas?  How large were the applied corrections?

l. 915: is "DN$_A$" equivalent to analog-to-digital 'counts'?  If so, I suggest using 'counts' since it's a more common term.

5. Appendix A:

   a. 'Measured Quantities': multigas: correct delta $^{13}$C notation
   b. Correct display of Size (L x W x H)
   c. I'm not sure I understand: how is the accuracy of the DOAS (1 ppm*m) smaller than its precision (5 ppm*m)?
   d. What is the model of $CO_2$ sensor that was used? I tried to check the listed parameters against the manufacturer specifications but couldn't do so with confidence because the model number of the multigas $CO_2$ sensor isn't given. What I did find suggests that the listed precision and accuracy are not supported by the available documentation. This is another place where open data would help substantiate the authors' claims. Based on what I can find, the *digital resolution* for the lowest-range smartGAS $CO_2$ sensor (F3-212205-05000) is listed as 1 ppm (the spec listed in the authors' table), but the manufacturer-specified '3σ detection limit' is listed as either ≤ 8 ppm or ≤ 20 ppm, depending on how it is configured ('standard' or 'fast'; I'm not familiar with these sensors to know what this actually means). These specs suggest that the random noise is more like ~3 to 7 ppmv at 1σ. Is this correct? Was the $CO_2$ sensor modified somehow to improve its precision? How were the values given in the table derived? Furthermore, Figure 3 indicates that the $CO_2$ signal is recorded using the Arduino's 10 bit ADC, so the best resolution possible will be ~1 ppm based solely on the ADC bit depth and sensor range (listed as 0-1000 ppm), assuming the full analog range is utilized.

   Please revisit these and other specifications in the table and clarify their meaning (1σ, 2σ, etc) and how they were determined or where they come from.

   https://www.smartgas.eu/fileadmin/11_aktuelle_datenbl%C3%A4tter_flow/DS_F3-212205-05000_CO2_2000_ppm.pdf
   e. The radio link is listed in the Appendix as 400 MHz whereas a 900 MHz radio is specified in line 186. Which is it? Are these bands legal in PNG?  What models of radios were used?
   f. Please specify the volume of the tedlar bags (presently indicated as "4 Tedlar bags (X L))"

6. Appendix B: Why are $H_2O$ mixing ratios calculated on a dry basis (Eq B2) and other gases calculated on a wet basis (Eq B1)?  The P correction on the sulfur sensors will be very small, but in formal terms it would be better to be consistent.  Rarely are such details given in volcano-gas papers, so even with the discrepancy I am happy to see these equations laid out.

**Editorial Suggestions and typos**

1. Correct all instances of 'in-situ' to 'in situ' (no hyphen, no italics), in line with EGU and other common editorial style guides.  See EGU's section on English guidelines and house standards: https://publications.copernicus.org/for_authors/manuscript_preparation.html

l.925 (**E**quation B4)

**References**

Brown, S.K., Jenkins, S.F., Sparks, R.S.J., Odbert, H., Auker, M.R., 2017. Volcanic fatalities database: analysis of volcanic threat with distance and victim classification. J. Appl. Volcanol. 6, 15. https://doi.org/10.1186/s13617-017-0067-4

Kelly, P.J., Kern, C., Roberts, T.J., Lopez, T., Werner, C., Aiuppa, A., 2013. Rapid chemical evolution of tropospheric volcanic emissions from Redoubt Volcano, Alaska, based on observations of ozone and halogen-containing gases. J. Volcanol. Geotherm. Res. 259, 317–333. https://doi.org/10.1016/J.JVOLGEORES.2012.04.023

Liu, E.J., Aiuppa, A., Alan, A., Arellano, S., Bitetto, M., Bobrowski, N., Carn, S., Clarke, R., Corrales, E., de Moor, J.M., Diaz, J.A., Edmonds, M., Fischer, T.P., Freer, J., Fricke, G.M., Galle, B., Gerdes, G., Giudice, G., Gutmann, A., Hayer, C., Itikarai, I., Jones, J., Mason, E., McCormick Kilbride, B.T., Mulina, K., Nowicki, S., Rahilly, K., Richardson, T., Rüdiger, J., Schipper, C.I., Watson, I.M., Wood, K., 2020. Aerial strategies advance volcanic gas measurements at inaccessible, strongly degassing volcanoes. Sci. Adv. 6. https://doi.org/10.1126/sciadv.abb9103

Roberts, T.J., Saffell, J.R., Oppenheimer, C., Lurton, T., 2014. Electrochemical sensors applied to pollution monitoring: Measurement error and gas ratio bias — A volcano plume case study. J. Volcanol. Geotherm. Res. 281, 85–96. https://doi.org/10.1016/j.jvolgeores.2014.02.023

Rüdiger, J., Tirpitz, J.-L., Maarten De Moor, J., Bobrowski, N., Gutmann, A., Liuzzo, M., Ibarra, M., Hoffmann, T., 2018. Implementation of electrochemical, optical and denuder-based sensors and sampling techniques on UAV for volcanic gas measurements: examples from Masaya, Turrialba and Stromboli volcanoes. Atmos. Meas. Tech 11, 2441–2457. https://doi.org/10.5194/amt-11-2441-2018

Syahbana, D.K., Kasbani, K., Suantika, G., Prambada, O., Andreas, A.S., Saing, U.B., Kunrat, S.L., Andreastuti, S., Martanto, M., Kriswati, E., Suparman, Y., Humaida, H., Ogburn, S., Kelly, P.J., Wellik, J., Wright, H.M.N., Pesicek, J.D., Wessels, R., Kern, C., Lisowski, M., Diefenbach, A., Poland, M., Beauducel, F., Pallister, J., Vaughan, R.G., Lowenstern, J.B., 2019. The 2017–19 activity at Mount Agung in Bali (Indonesia): Intense unrest, monitoring, crisis response, evacuation, and eruption. Sci. Rep. 9, 8848. https://doi.org/10.1038/s41598-019-45295-9

---

## Referee Comment (RC2) · Anonymous Referee #2 · 26 Jan 2021

The manuscript presents a multi-rotor aerial drone system equipped with different gas analysing and gas sampling instruments to investigate the composition and flux of volcanic plumes. The drone can reach altitudes of 2.000 m above take-off level, and ranges in the order of 5 km; these are quite remarkable specifications for a vertical take-off and landing drone with a payload mass of up to 2 kg and a take-off weight of max. 6 kg. During a field campaign in May 2019, the plume of Manam volcano in Papua New Guinea has been comprehensively characterized by measuring the in-situ concentrations of $SO_2$, $CO_2$, and $H_2S$ in the plume, calculating the total $SO_2$ emission rate by taking into account the determined plume speed data, and by taking gas samples from the volcanic plume with post-flight analysis on the ground for halogens

and carbon isotopes. The data obtained with the drone system have been compared with additional data from ground-based and aerial measurements as well as with atmospheric model calculations and are - as far as such additional data were available - in good agreement. Similar volcanic plume measurements by using multi-rotor drones have already been published, see inter alia the cited references Stix et al. (2018) and de Moor et al. (2019). In the present manuscript, the special aspect is the versatility and modularity of the multi-rotor drone system used. The promising applications of vertical take-off and landing drones in the field of volcano research and monitoring are illustrated.

Page 1, line 22: Instead of "…multi-copter drone…" it seems more appropriate to use the term "…multicopter drone…" or – as in the title of the manuscript – "…multi-rotor drone…".

Page 2, line 56: Instead of "…Mori (2016)…" it should read "…Mori et al. (2016)…". Instead of "…multi-rotor…" it seems more appropriate to use the term "…multi-rotor drone…" consistently – also in several other sections.

Page 4, line 150: At first glance, the presented finding that a balance of "rise and forward motion" is more favourable in terms of energy consumption than "moving only in one direction at a time" seems to be obvious. But possibly a rule for the optimal balance between "rise and forward motion" in terms of minimum energy consumption has been identified. If applicable, this should be specified.

Page 5, line 179: Instead of "longer propellers" it seems more appropriate to use the term "larger propellers" or "propellers with a larger diameter" – also page 24, line 659

Page 7, line 217 – drone drift method: Even if there is no side wind, a multi-rotor drone drifts slightly in one direction when GPS lock mode is deactivated. Has it been investigated how large this offset is and was this taken into account when measuring the plume speed?

Does the drift speed determined from the GPS data also include a vertical speed component? Or does the drone only drift in a lateral direction and maintain the position in the vertical?

Page 7, line 222 – onboard anemometer: According to Appendix A, an FT205EV anemometer has been applied onboard the drone. Please specify whether this anemometer measures only the horizontal or also the vertical component of the wind speed?

Has it been investigated whether - and if so to what extent - the wind measurement using the anemometer mounted on top of the multi-rotor drone was influenced by the air flow created by the propellers?

In addition to the photo in Fig. 1, it would be useful to have a sketch showing the exact location of the onboard anemometer and in particular its horizontal and vertical distances from the propellers.

Page 8, line 245: Please check "...described in section 2.3.1" since this section does not seem to exist.

Page 11, line 307: The wording "...using homemade software..." is ambiguous, as it could be understood to mean that the software is developed by a software provider with the name "homemade" (which exists); contrary to that it might be intended to indicate that the software is "self-developed". Please clarify if necessary.

Page 11, line 316: Instead of "...a rapidly fluctuation signal is measured..." it should read "...a rapidly fluctuating signal is measured...".

Page 11, line 325: The wording "...a practical solution is to take a sample of a time-varying signal and then expose the sensors to the sampled gas..." is possibly inappropriate. Maybe what is meant is that "a sample of gas is taken" and then the sensors are exposed to this sample of gas. Please consider this and amend the wording correspondingly, if necessary.

Page 11, line 326: It is stated that "Our system fulfills these two criteria: the sensors have similar response characteristics. . ." while on page 12, line 345 it is stated that "Because our sensors operate according to different principles, the sensor response times are usually different;. . .". Please clarify whether different sensors are meant in each case.

Page 12, line 359: It is stated that "Such dynamic changes (with frequency components of higher than 0.5 Hz) in plume composition are assumed to be improbable for most typical scenarios". Are there any published studies or own measurement results on this subject?

Page 15, line 444 – small rotary pump: Some of these small rotary pumps have vanes made of graphite, which can cause carbonaceous abrasion. Has it been investigated whether using such a pump influences the gas composition and isotopic analysis?

Page 18, line 530: Please indicate whether the plume speed measured using the anemometer when "the drone is kept in a fixed position" is the horizontal plume speed component only or the sum of horizontal and vertical plume speed components, i.e. including the buoyance of the plume.

Page 19, line 538: Please indicate whether the plume speed measured using the drone drift method is the horizontal plume speed component only or the sum of horizontal and vertical plume speed components, i.e. including the buoyance of the plume.

Page 20, line 556: Please clarify that the altitude is "1000 m AMSL".

Page 20, line 561: A reference is missing in the caption of Fig. 10.

Page 22, line 590: A reference is missing in the caption of Fig. 12.

Page 24, line 653: Please correct ". . .of of. . .".

Page 24, line 655: Please correct ". . .the the. . .".

Page 25, line 677: The trajectories show remarkably long flight distances in both horizontal and vertical directions, especially considering the relatively small drone size. Please indicate whether the drone was manually controlled only during these flight distances and, if so, whether there was any support for the pilot, for example through onboard cameras.

Was the multi-rotor drone also flown occasionally through a volcanic ash cloud? If so, did this have any negative impact on the measuring instruments or the drone, e.g. wear on the rotor blades of the drone?

Page 26, line 718: Has the cited and listed reference "ARELLANO et al. (2016)" already been published or is it otherwise available online?
* * *

---

## Author Comment (AC1) · 6 Mar 2021

Below we have listed first the comment/question written by the reviewer, followed by our response.

Reviewer: This article describes a small, electric-powered multi-rotor drone and several payloads that were used for volcanic gas sensing and sampling at volcanoes in Papua New Guinea between 2016-2019. The authors focus on technical descriptions of the payloads (DOAS, multi-GAS, a denuder system, and gas- bag collection system) and modifications that were made to the drone platform to improve its endurance. This contribution appears to serve as a technical companion paper to (Liu et al., 2020), who

Interactive
comment

discuss the volcanologic significance of the obtained gas composition and emission rate results from the 2019 campaign

Response: Yes, the paper can be regarded as a technical companion paper to (Liu et al., 2020). It is a technical paper focused in only one of the platforms used in Manam, one that was capable of measuring all target parameters of the field campaign. Although we have a comprehensive dataset to determine emission rates for SO2 and CO2, as well as several other gases, the dataset is quite limited. As the data was gathered under a larger campaign with several other instruments and techniques we feel that the combined data from all these measurements better represent the parameters studied, as given in (Liu et al., 2020)

Reviewer: Some of the payloads used in the experiments have been described previously (e.g. the DOAS system, denuder system, and 'Sunkist' instrument; Rüdiger et al., 2018), but the manuscript does include descriptions of a new multi-GAS unit developed by Chalmers U. that includes the innovative integration of a mini anemometer to obtain windspeeds, as well as a plume sampling unit for collecting bagged samples for posterior carbon-isotope analysis. To me, the most novel aspect of the manuscript is the presentation and analysis of the two methods for determining plume speed; most of the other instruments and techniques have been in use for some time. Accurately determining plume speeds is critical for determining volcanic gas emission rates, and the instrument and methods comparison shown here are helpful for addressing this important issue .

Response: The merit of this paper is not that the used drone can reach exceptional range or altitude, nor has significantly new instrumentation. Although some of the systems have been described previously (denuder and Sunkist), we also present independent developments for the MobileDOAS and MultiGAS that allow real-time measurements, as well as a new method to correct for time-response differences of MultiGAS sensors. The main purpose is to demonstrate that it is now possible to go to a very remote and inaccessible volcano, with almost no infrastructure, and launch a set of different payloads to a distant (5 km) gas plume at 2000 m height and perform a unique set of measurements over 5 days with only 2 persons.

Reviewer: The technical emphasis of the manuscript is appropriate for Atmospheric Measurement Techniques and the operational 'lessons learned' will be valuable and of interest to the volcanic gas community. The manuscript is generally well-written and structured but there are some items that need to be addressed prior to publication. Broadly, my main concerns (documented below) are that the manuscript is too vague in places, and that supporting data are incomplete, contain mistakes, or are not available in an open repository. The scientific value of the collected gas measurements are hardly discussed (perhaps a little more effort could be made here, or would it overlap too much with Liu et al.?), therefore I feel that the technical contribution must be significant and substantive to warrant publication. These issues compromise the study's impact and value in its present form but should not be too difficult to remedy. The article will be appropriate for publication in AMT after these issues and the comments below are resolved. I hope that these comments are helpful

Response: We respond to each of the issues raised in the following paragraphs.

Reviewer: Data availability At present the manuscript does not adhere to the data standards for AMT. Line 901: 'The datasets generated for this study can be provided upon request sent to the "Corresponding Author".' The data from this study needs be made open and accessible, in accord with current community and journal standards. AMT/EGU Data policy: https://www.atmospheric-measurement-techniques.net/policies/data_policy.html Unfortunately, the data that are available appear to be incomplete and contain mistakes. For example, I tried to further examine the multiGAS data presented in the study but encountered significant difficulties in attempting to do so, as described below.

MultiGAS data from the experiment are said to be available from Liu et al., 2020 (l. 580-581): https://advances.sciencemag.org/content/suppl/2020/10/26/6.44.eabb9103.DC1

[Figure]

Data that I was able to find in the supplement to Liu et al. (2020) includes data from three flights on May 22 and May 23, 2019 but not from other dates (for example, data from 2016 are plotted in Figure 4 and data from May 26 are plotted in Figure 11 and are not in the Liu et al. supplement). The data available in the supplement apparently include data from two multigas instruments: one from U. Palermo and the Chalmers instrument described herein (although the supplement does not readily indicate which data came from which instrument, or I missed the explanation somewhere – my apologies if I simply missed it!). The supplementary data also do not include absolute timestamps (only sequential integer seconds) so it's not possible to precisely connect these data to the results listed in the Liu et al. study by date/time, and while they do include lat/long, the altitudes are missing which makes plotting and understanding the flight paths hard. Since Liu et al. (2020) emphasize the data from the U. Palermo instrument, my best guess is that the first two tabs include data from that instrument and my hunch is that the third dataset (Raw data 23-05-19 B) came from the Chalmers instrument, but of this I am not positive. My hunch that the presented data come from different instruments is supported by the observation that the data formats are different (e.g. the first two data tabs have lat/long listed as decimal degrees and the third has lat/long as UTM), but I really don't know for certain. If my hunch is correct, then what I take to be Chalmers multiGAS data from May 23 appear to be very poor, and only show two $SO_2$ peaks near the end of the file (one of which is partially truncated; shown below) and $CO_2$ is poorly correlated with $SO_2$. The poor correlation between the $CO_2$ and $SO_2$ would appear to violate assumption 2 of the analysis routine (l. 383). Elsewhere the $CO_2$ data show large apparent jumps of $\sim$15 ppm; is this some kind of interference? Finally I note that no units are given for any of the measurements, and the air temperature (Tair) appears to be listed in A/D counts(?) rather than sensible units (the 'Tair' values range from 5369 to 6805). I could not find data from the Sunkist unit anywhere.

Response: Thanks for pointing out this major issue with the present version of the manuscript. The observation led us to revise the supplements in Liu et al. (2020)

and found that only data from the Palermo instrument has been presented there. All comments about the assumption of Reviewer #1, including the figure, are therefore not relevant for the data concerning this manuscript. The following comment has been made by Emma Liu: "I would like to add a couple of points on this topic. First, I agree that more explanation in the form of a readme file would have been helpful to the interpretation of the supplementary files. Second, the data given in the supplement are the raw data prior to sensor response correction to CO2 that accounted for the internal averaging. The response correction applied to the Palermo instrument did not assume correlation, and was instead based on lab tests and modelling. Third, the third tab in the supplement 23-05-19-B, which the reviewer refers to, was the flight in which the fixed-wing drone was lost. • This is why the time series is abruptly truncated part way through a peak. • The log files are not as complete as we would have liked for this flight, as we could not retrieve the onboard SD card. Only a subset of the data was sent to the ground-station in real-time as lower resolution files that could be retrieved later, RH and T were not correctly transmitted. This is why the data format is different. I should have noted this explicitly in the supplement. • The small, transient jumps in CO2 that are mentioned are attributed to radio interference, as we have had issues with this in the past."

To fulfill the requirements raised by Reviewer #1, we have prepared and Excel file containing all data presented in this manuscript. This Excel file is added as a supplement.

Reviewer: Specific Comments 1. The range achieved in the present work ($\sim$ 5 km) is good for a multirotor system but is not especially noteworthy and has been achieved previously with commercial multi-rotor drone systems, as summarized in James et al., 2020, Table 2 (copied below) and the references therein (quote from James et al.): "UAS equipped with miniaturised gas sensing instrumentation (see Section 2.2.3; Figures 2A, 15) are now bridging the gap between direct measurements and remote sensing observations, enabling repeatable, proximal measurements from ranges >5 km [Pieri et al., 2013a; Shinohara, 2013; Diaz et al., 2015; Mori et al., 2016; Xi et al.,

2016; Di Stefano et al., 2018; Rudiger et al., 2018; Stix et al., 2018b; Kazahaya et al., 2019; Liu et al., 2019; Schellenberg et al., 2019; Syahbana et al., 2019]." Some of the UAS listed in James et al. are gas-powered, but, for example, Syahbana et al., 2019 used a small electric-powered fixed-wing drone that launched from 11 km distance and gained > 3000 m altitude to monitor Agung Volcano during unrest and eruption. Endurance to launch from >10 km is significant because the vast majority of deaths from PDCs at explosive arc volcanoes occur within ≤10 km distance from the vent (Brown et al., 2017). The introduction should reflect the fact that small, electric, multi-rotor drones cannot match the range possible with equivalent fixed-wing units, but they do have significant advantages in terms of maneuverability, ease of take-offs and landings, ease of integrating payloads, etc., as discussed in the manuscript.

Response: Our goal is to develop a system that can be used to study the large set of active volcanoes that are not under explosive eruption but where the summit region is still inaccessible due to high risk or complicated logistics. For many of these volcanoes 5 km distance and 2 km altitude is enough to be able to do measurements of the plume. We have learned from long experience that although the main wind direction can be well defined the actual position of the plume center may move around significantly within a short time-span. Thus to be able to study these plumes we need a drone with enough capacity to fly to a plume 2000 m above the launch site and 5 km away, maneuver to the plume center and measure for 5 minutes or more and then safely return. Another advantage of multi-rotor drones is the possibility of conducting bag sampling that is not yet demonstrated for fixed-wing platforms. We also notice that from the examples presented in James et al. (2020) review paper, multi-rotor drones are claimed to have those ranges, but in practice only the system used by Mori et al. (2016) has reached similar ranges of altitude and distance as our drone did during several field campaigns in Papua New Guinea.

Reviewer: The notion of what constitutes 'long range' and 'high-altitude' is subjective, but given that the drone's performance more or less matches the capabilities of other

similar systems it may be appropriate to consider changing the title to the present work to "A multi-purpose, multi-rotor drone system for volcanic gas plume measurements ".

Response: Although a fixed wing drone can easily achieve the range and altitude required, the maneuverability required to find and stay in the plume is lacking. Many of the drones referred to in the attached Table 2 also have a considerable total weight. This hamper the transport to suitable launch sites and may exceed limits set up in local regulations for UAV flights (7 kg is a limit in Sweden). Also, battery size is a limitation both due to air transport regulations and charging conditions in the field. Considering all this we believe that the multi-rotor drone with less than 7 kg total weight that we present here is a good compromise between range, maneuverability, and size.

Reviewer: 2. Figure 1 of the manuscript is copied verbatim from Liu et al. (2020) supplement Figure S3b (shown below) and without correct attribution. Although the team for the two papers are much the same, 'recycling' figures in this way is poor practice and - at minimum - any previously published images and/or figures need to be cited properly.

Response: We have replaced the figure using a different background photograph and adding more details for the sensors and other instrumentation (New Figure 2, attached as Fig 2 here). Caption: "Photo of the multi-rotor drone with modular payloads. The MultiGAS unit includes in situ sensors for gas composition (XA-denotes concentration of species A, p-pressure, T, temperature, %RH-relative humidity, xyz-tilt coordinates), a gas-sampling unit and an anemometer. The MobileDOAS is used for remote sensing of gas flux. The modules are clamped to the drone at balanced position. The battery pack is placed below the drone chassis to lower the center of gravity of the system. Flight and sensor data are telemetered in real-time (photo courtesy of Matthew Wordell)".

Reviewer: l.149-156: These seem like very important operational observations. However, critical details are left out or are too vague. "The drone's angle. . .also proved to be of great important for energy consumption." Can this be made more specific and

actionable? What is an optimal flight vector? Roughly how much endurance might the flight optimization gain? Why did power consumption go up so much when encountering clouds ?

Response: The energy consumption probably goes up in clouds due to increased turbulence, forcing the drone to react. More on this is given on the new page 7 and new Fig 1. , attached as Fig 2 here with new caption "Flight data from the drone flight shown in Figure 9. The upper panels show time-series of yaw, pitch and roll angles, the thrust (percentage) and the altitude of the drone. Notice the high variability of the parameters associated with acceleration, hoovering and interference from clouds."

Reviewer: l. 166-168: "It was found that the time needed for switching between different payloads could be considerably reduced by changes in the drone frame and payload designs (balance, power connection, data access, telemetry)." The present description is too vague. What 'changes' were made? How were instruments mounted? How was balance checked ?

Response: We have added: "The batteries were mounted under the frame and could be changed with a "click" locking. This enabled fast switching of batteries and improved the balance especially at take-off and landing. The payloads were mounted on individual plates that was locked in place on the drone platform with a "click" lock. This enabled the payloads to be pre-balanced and no further balancing of the drone was needed after replacement of payload. A special connector on the drone gave the payloads power and access to the drone mounted telemetry. This power connector was always turned on to make it possible to do pre-flight and post-flight operations on the payload instrument without turning on the main drone electricity to save power".

Reviewer: l. 169-171: "Access to the drone flight logs were found to be useful. . ." I agree that they would be very useful for the reasons stated. Please make the logs available as part of the data release.

Response: We think it is a bit too much to include the full flight logs for all the flights in

the data release. We have however included some flight log data for a typical flight in the new Fig.1. (attached here as Fig. 1.)

Reviewer: l.284-296: On my first reading it was unclear if the $CO_2$, $SO_2$, and $H_2S$ sensors were self-made or commercially-available units. Only upon reading Appendix A it became clear that the sensors were not self-made. Please amend the section to include specifics of the sensors so that readers don't need to consult the Appendix to find this. Also, the descriptions of the NDIR and electrochemical sensors can be shortened since their measurement principles are well known and described in the manufacturers' documentation .

Response: Since we don't indicate that sensors were self-made we do not think this will be a cause of confusion for most readers. But we have added a sentence at the end of this subsection: "and technical specifications of the sensors are given in the Appendix" to avoid any misunderstanding. We would like to keep the brief description of the principles of operation of the MultiGAS sensors for two reasons: first, because this introduction is needed to understand the reasons why the time responses may be different, and what changes can be done to make them more equal; and second, because AMT is a journal mostly read by the atmospheric science community and some of these sensors are not as widely known by scientists other than volcano geochemists.

Reviewer: Figure 3: the box diagram shows the anemometer located inside the instrument enclosure

Response: This has been corrected

Reviewer: 3. Measurement of in-plume $H_2O$ mixing ratio l.299-301 If I understand correctly, the RH sensor included in the multigas unit measures RH inside the instrument box, not in the sample gas stream. If true, the instrument therefore does not achieve its claimed capability of measuring in-plume $H_2O$ (l. 26). The authors seem to implicitly acknowledge this detail in the conclusions (l. 669) where they list $CO_2$, $SO_2$, and $H_2S$ as measured gases instead of $H_2O$, $CO_2$, $SO_2$, $H_2S$ as claimed earlier and in

the abstract. It's confusing to me why the authors go on to discuss how to calculate in-plume $H_2O$ mixing ratios when their system doesn't actually have that capability: l. 299-301 "Our system, however, measures these variables only inside the instrument box, so the mixing ratio is representative of ambient gas passively diffusing in the interior of the unit.." I'm very sorry if I've misunderstood something but I've read through lines 297-303 several times now, and the description and Figure 3 indicate that T, P, and RH measure conditions inside the sampling box, not in the sample stream or in ambient air outside the instrument enclosure. While it's perhaps acceptable to use the P record as 'ambient' or 'near-ambient' P, the T and RH will be useful as diagnostics, but not as plume or ambient measurements. As presented, this is very confusing. I recommend clarifying which in-plume measurements the system supports, and which measurements are for diagnostics. Since the RH appears to be intended for diagnostics, then there is really no need to discuss conversion to in-plume $H_2O$ mixing ratio and that text can be deleted .

Response: We have omitted the mention to plume-$H_2O$ as a measurement target in the abstract. The entire sentence referred by Reviewer #1 reads: "For the case of $H_2O$, the mixing ratio can be derived from measured relative humidity, pressure and temperature, following known thermodynamic laws (see Appendix B). If the measurement of such variables is done inside the sampling circuit, the $H_2O$ mixing ratio of the sample can be determined simultaneously to the other species. Our system, however, measures these variables only inside the instrument box, so the mixing ratio is representative of ambient gas passively diffusing in the interior of the unit; $H_2O$ therefore varies differently than the other species as it is determined from outside of the closed system". We don't see any reason for misinterpretation in this paragraph. We acknowledge that our system is not tailored for measurement of $H_2O$ under the same conditions as for the other species. But we include a hint on how this could be implemented. However, we notice from experience that it is harder to distinguish the volcanic signal of water from the background, specially in the tropics. This is because ambient water vapor and water from shallow hydrothermal systems produce a highly variable background signal.

Reviewer: Furthermore, the desired resolution of the water measurement is purported to be 1 ppmv (l. 110) - an ambitious goal, to be sure - but the precision of the RH sensor is stated as 3% RH. The point is moot since it appears that the instrument was not designed to measure plume H2O, but for the sake of argument let's say the total P = 1000 hPa, the PH2O is 20 hPa, and saturated vapor pressure is 25 hPa. In this case the RH would be approximately 80%. Here, ±3% RH precision would translate to about ±0.75 hPa or about 750 ppmv (it's unclear if the given precision is 1$\sigma$ or a range, etc., please clarify here and throughout). This example suggests that in practical terms it would be impossible to achieve 1 ppmv H2O resolution with the specified sensors. In addition to random and systematic errors on the RH measurement, I would expect some error in the relationship used to convert RH to mixing ratio, random and systematic errors in the needed P+T measurements, etc .

Response: The only mention to 1 ppmv as a target is when defining the goals that motivated the development of our system, thinking mostly on SO2 and CO2. But the sentence indeed included H2O and we have now corrected this error. We have changed the phrase to "a few ppm" to avoid strict adherence to a strict and arbitrary detection limit. The rest of the paper shows the actual capabilities of the instrument.

Reviewer: Please carefully edit the manuscript so that the measurements made are characterized accurately, and that realistic analytical values are given as design goals, and that the methods used to characterize the accuracy and precision of the various sensors are described and/or listed in the measurement specifications. For example, the denuder section (2.6) gives a clear statement on how LOD and LOQ were calculated (l. 501-503 ).

Response: We think we have included enough detail about the specifications of the sensors in Appendix 1, chiefly model numbers that anyone can check on the manufacturers' websites. Details about calibration are given below.

Reviewer: l.326-327: Please include the time constants of the sensors here. Are the

values listed in the appendix t90? Please clarify .

Response: Yes, the manufacturers specification correspond to t90, which is now specified in Table 1.

Reviewer: L.327-329: This is an interesting idea, but I do have concerns about its viability. Would a ~1L tedlar bag provide enough gas to get a good 'plateau value' from the sensors? How long do the sensors take to plateau during calibrations? Also, at low concentrations I would expect some sorption of S-containing species that could impact the results.

Response: We have added: "In this mode the gas from the teflon bag is circulated through the detectors in a closed loop and thereby exposing the detectors for the constant gas concentration in the sample for several minutes. Another advantage here is that any possible losses, i.e. wall effects, could be monitored and compensated for. This method was tested only once in the actual field campaign, because the limited gas samples was instead used for isotopic composition analyses." The only test made to practice the method used a sample that was too diluted for a successful measurement of $CO_2$, but it probed that the closed-loop principle could work. The figure of the uncalibrated signals for this test is shown as Fig.3. here (measurement on 22 May 2019). As clearly shown, the $SO_2$ signal is detectable and its curve of growth stabilized in about 30 s (t90), which matches well with the manufacturer's specifications. But the signal, after calibration, is less than 1 ppm and therefore the $CO_2$ signal above background is below the detection limit of the instrument (remember the molar ratio for both species at Manam was found to be close to 1). The figure also shows a noise picked up by the $CO_2$ sensor, which we attribute to the radio. This signal is subtracted together with the background as part of the corrections. This information is included in the Supplement.

Reviewer: l.337: what is "the time of variability in gas concentration"? The meaning is not clear in the explanation .

Response: We provide a detailed explanation in the paragraph that follows: "The first

characteristic is determined by variability in emission, variability caused by local turbulence at the point of measurement and variability caused by relative transit of the drone with respect to the plume" In other words, this is time characterizes the variability of the 'true' signal, i.e. of the signal that would be measured by an hypothetical perfect instrument reacting instantaneously to the measured signal and sampling at infinite (or much higher than Nyquist's) rate.

Reviewer: l.360-361: 20 seconds to exchange the volume of the CO2 optical cell seems like an awfully long time. Can the pump rate be increased to shorten this time? It would be very useful to see what a step function looks like during calibration of the CO2, SO2, and H2S sensors .

Response: The calibration curves for CO2 and SO2 sensors are presented in the Supplement.

Reviewer: 4. Multigas data processing technique l.365-366: "...only one energy storing and one energy dissipating component..." I think what's being referred to here is a resistive-capacitive circuit (RC), which are classically described as first order systems. Perhaps consider recasting this section using more standard terminology .

Response: Thanks for the suggestion. However, we prefer to keep it as it is because it corresponds to a generic description of first-order systems, not a specific realization that uses only a resistive and a capacitive component.

Reviewer: Equation 1: this looks like an interesting approach; amplifying the signal to better approximate the input versus instead of lowpassing a 'fast' sensor to match a slower sensor. I couldn't get a copy of the reference within the timeframe of this review. Are there other more easily-accessed references that explain this theoretical approach? Is ïĄ́ť the first order time constant here? Most of the electrochemical sensors I use have ïĄ́ť values between about 2 and 6, so would a1 normally be 2 to 6 and a0 = 1 ?

Response: The reference is a report from a field campaign in 2016, which is free to access through this link: https://research.chalmers.se/publication/254380 However, there is probably not enough detail in this report to fully implement the method. To simplify this, we have shared the essential steps of the code in the Appendix. According t Eq. 1, the characteristic time  = a1/a0, and the sensitivity equal to 1/a0. If only the value of the characteristic time is known, one needs to provide the value of the sensitivity (ppm/mV or similar, reciprocal of a0) to infer the value of a1

Reviewer: l.379: I appreciate that the equations for the frequency-dependent amplitude and phase lag are given, but they could use a little more explanation for readers and a reference. At minimum, the angular velocity () should be defined and it should be made clear that these relationships describe the phase shift in radians (I suppose that tan-1()*180/pi could be given if degrees are preferred ).

Response: The reference (R. Pallas-Areny and J. G. Webster, 1991, Sensors and Signal Conditioning, Wiley, New York) has been added. The definition of angular frequency (2*pi*f) is well known.

Reviewer: l.382: Please provide evidence that shows the responses of your sensors (e.g. many studies show calibration peaks ).

Response: Please see Supplement S3.

Reviewer: l.383: I am concerned that the assumption that the input signals are highly correlated is dangerous; this may work for measurements with high signal to noise and for homogenous plumes, but heterogenous plumes exist and sometimes ambient background variations can be significant (e.g. Kelly et al., 2013). This is a significant weakness of the outlined approach .

Response: We discuss this in the manuscript: "Sampling a heterogeneous mixture would produce different ratios at different times, complicating both the measurement and the interpretation of the results. In volcanic emissions, drastic changes in molar

ratios within minutes are unlikely if the gases come from the same source. But if the plume mixes emissions from different vents or if large local heterogeneities affecting unequally the chemistry or condensation of different species (e.g. for plumes with heterogeneous concentration of ash), changes in gas molar ratios can occur even on short time scales". We doubt this presents a serious limitation of our method. For one, because the drone-based or a ground-based MultiGAS instrument is sampling a rather limited volume of the plume/fumarole (not an entire plume where heterogeneities in relative composition may exist). For another, because the method is applied on a time window of a few minutes, during which drastic changes in the ratios in the measurement spot are highly unlikely. And finally, because if there would indeed be a drastic change in ratios, the method would still find a correlation, but one with a coefficient much lower than 1, giving a method to identify such drastic changes by the poor correlation found between the signals.

Reviewer: l.384-386: It's not clear to me what these assumptions mean. Is there another way to clarify ?

Response. We added a sentence: ", because the high variability in the signal is required for the cross-correlation analysis" The two conditions essentially mean that we need a signal that is long enough to warrantee that the sensors have been exposed for times longer than the exchange time of the cavity (CO2) and that there are fluctuations that would allow to make an effective cross-correlation between the two time-series. The actual times depend on the signal and the sensors, but a few minutes for measurements close to a turbulent fumarole would be enough.

Reviewer: l.397-405: A worked example is needed here, perhaps as part of the data release. It's not enough to say that the evaluation happens in matlab. If the intent is for others to try to use this technique, then an example dataset would be most useful. When iterating the time response factors, is only one parameter varied (a1) and a0 set to 1 ?

Response: Please refer to Appendix B. We have added a sentence pointing the reader to the implementation found there.

Reviewer: Figure 4: I'm surprised the CO2 shows so much variation with the 20 second time necessary to exchange the gas in the optical cell. Also the response time in the Appendix is listed as 20 seconds for the SO2 and 30 seconds for the CO2, yet the CO2 appears to be much 'faster' than the SO2 sensor. How is it possible that the CO2 sensor has such a long exchange time, slower response, and yet shows much sharper measurements than the SO2? The CO2 also shows much smaller corrections than the SO2 sensor. Can this be explained and clarified or corrected? What values of a0 and a1 delivered the optimal fit ?

Response: This is because we are showing measurements taken with the Sunkist in 2016 on the central vent of Tavurvur volcano. We have added a sentence to make this clearer. The Sunkist sensors are different and have very different time responses. This example is taken to illustrate how to use the method because the MultiGAS used in Manam have sensors with almost equal response times and the signals in the plume have fewer wild fluctuations.

Reviewer: The corrected signal in 4b still shows considerable scatter (as do the data in Figure 11), which suggests that perhaps the model isn't working so well. It would be useful to compare the results of this method to the approach described by Roberts et al. (2014 ).

Response: It would indeed be interesting to compare both methods, but we think this could be done in another study. To include it here would scatter too much the focus of the article, where the time-response correction method is only a small part concerning of one of the multiple payloads. However, we can notice that the method proposed by Roberts et al. (2014) relies heavily on parameters derived from laboratory calibration and implicitly assumes that the time responses of the sensors will not be affected by the measurements in the field. While this may well be true, especially for well-designed

instruments and robust sensors, there is a risk that measurements at very extreme conditions (high temperature close to fumaroles or low pressure at elevated plumes) have an impact on the dynamical response of the sensors. The method we propose is free from these problems because it optimizes the parameters governing the dynamics of the sensors for the actual measurement conditions. The method we propose does not need laboratory calibration and it is easy to implement.

Reviewer: l.416-420: I disagree that characterizing sensor responses is overly time-consuming. Characterizing such responses offers many advantages for tracking sensor health and provides a basis for simple, reproducible, and automated data processing routines that require no assumptions of plume homogeneity like the presented method. What evidence exists that lab-derived time responses differ from field performances? This section feels like an overreach and the claims should be substantiated or revised .

Response: Characterization of sensor responses may not be overly time-consuming when the logistical conditions allow to visit a well-equipped lab. Doing this in a remote island of the Pacific with basic infrastructure is quite a different story. The assumption of plume homogeneity has been discussed above, but if there is heterogeneity, time response characterization in the lab will not help, because the ratios will be changing from time to time and no determination of a single ratio would be possible. Our method at least could signal the occurrence of such unusual sample. We recognize we are not presenting evidence for changes in time response between lab and field conditions, but this is only presented as a potential limitation of a method that relies only on lab calibrations. What we know, from the physics of the electrochemical sensors, is that resistances and capacitances used in the circuits are sensitive to changes in temperature, and if they change the response times of the sensors will also change. Perhaps this is not an issue for the measurements from a drone that are only exposed to the sample for a limited time, but it could be an issue for monitoring stations located close to high concentration fumaroles of elevated temperature for long periods of time. It is

precisely for this type of measurements that we think our method could be useful.

Reviewer: l.462: What was the value of the calibration gas? How large were the applied corrections ?

Response: Please refer to the Supplement

Reviewer: l. 915: is "DNA" equivalent to analog-to-digital 'counts'? If so, I suggest using 'counts' since it's a more common term .

Response: Done!

Reviewer: 5. Appendix A: a. 'Measured Quantities': multigas: correct delta 13C notation b. Correct display of Size (L x W x H) c. I'm not sure I understand: how is the accuracy of the DOAS (1 ppm*m) smaller than its precision (5 ppm*m )?

Response: Thanks for pointing this. The reported value for accuracy has been corrected. However, we think it is possible for accuracy to be lower than precision if we would adopt the definition of accuracy as the deviation from a value assumed to be true and precision as the standard deviation of a distribution of a number of measurements. One could find that the most probable value of the distribution of measurements has a deviation from the "true" value that is lower than the dispersion (standard deviation) of the distribution of measurements. In the way we defined accuracy we include both the precision and systematic sources of uncertainty, e.g. accuracy of absorptions cross-sections.

Reviewer: d. What is the model of CO2 sensor that was used? I tried to check the listed parameters against the manufacturer specifications but couldn't do so with confidence because the model number of the multigas CO2 sensor isn't given. What I did find suggests that the listed precision and accuracy are not supported by the available documentation. This is another place where open data would help substantiate the authors' claims. Based on what I can find, the digital resolution for the lowest-range smartGAS CO2 sensor (F3- 212205-05000) is listed as 1 ppm (the spec listed in the

authors' table), but the manufacturer-specified '3$\sigma$ detection limit' is listed as either $\leq 8$ ppm or $\leq 20$ ppm, depending on how it is configured ('standard' or 'fast'; I'm not familiar with these sensors to know what this actually means). These specs suggest that the random noise is more like ∼3 to 7 ppmv at 1$\sigma$. Is this correct? Was the CO2 sensor modified somehow to improve its precision? How were the values given in the table derived? Furthermore, Figure 3 indicates that the CO2 signal is recorded using the Arduino's 10 bit ADC, so the best resolution possible will be ∼1 ppm based solely on the ADC bit depth and sensor range (listed as 0-1000 ppm), assuming the full analog range is utilized .

Response: The model used was F3-212205-05000, this has been added to the Table. Accuracy is not the same as detection limit. The reported value for accuracy is estimated from the regression lines in the calibration curve at 1-sigma level. It is true that the Arduino uses a 10 bit ADC, but for the reason explained by the Reviewer, we used another ADC of 16 bits. Due to the low resolution of the ADC on Arduino Mega2560, which is 10 bits, the minimum voltage reading units is limited to 4.9 mV. For the measurement of the low concentration in the volcano application, a higher-precision ADC, the ADS1115, was used to improve the resolution to 16 bits (Xu, 2019). The specification of the ADC has been added. Thanks for thorough revision of this.

Reviewer: Please revisit these and other specifications in the table and clarify their meaning (1$\sigma$, 2 $\sigma$, etc) and how they were determined or where they come from .

Response: The parameters that represent dispersion are all given at 1-sigma level. This is standard use and we think the interested reader has been provided with enough information now (manufacturers model etc.) to double check this information.

Reviewer: e. The radio link is listed in the Appendix as 400 MHz whereas a 900 MHz radio is specified in line 186. Which is it? Are these bands legal in PNG? What models of radios were used ?

Response: We used two radios one at 900 MHz for the radio control of the drone

(replacing a standard of 2.4 GHz to achieve longer range), and one of 433 MHz for the payloads (wrongly stated before as 400 MHz, now corrected). This solution of independent radios was important to have autonomy and avoid saturation. Having the additional payload telemetry allowed us to retrieve the drone in a failed flight in 2018. Full permissions for flights and use of frequencies were granted by PNG Civil Aviation authorities for this campaign.

Reviewer: f. Please specify the volume of the tedlar bags (presently indicated as "4 Tedlar bags (X L ))"

Response: Corrected (1 L)

Reviewer: 6. Appendix B: Why are H2O mixing ratios calculated on a dry basis (Eq B2) and other gases calculated on a wet basis (Eq B1)? The P correction on the sulfur sensors will be very small, but in formal terms it would be better to be consistent. Rarely are such details given in volcano-gas papers, so even with the discrepancy I am happy to see these equations laid out .

Response: We were not aware of this discrepancy and adopted the correction for pressure suggested by the manufacturer as we were not able to perform characterization of the effect of pressure and temperature in the lab. Since the manufacturer's specification indicate a minor pressure effect, we think this method will not affect the results. We agree these details are usually not given and decided to include them now because of our own struggle trying to find how others have done it before.

Reviewer: Editorial Suggestions and typos 1...........Correct all instances of 'in-situ' to 'in situ' (no hyphen, no italics), in line with EGU and other common editorial style guides. See EGU's section on English guidelines and house standards: https://publications.copernicus.org/for_authors/manuscript_preparation.html

Response: Corrected!

Reviewer: l.925 (Equation B4 )

[Figure]

Response: Corrected!

Response: References We have added 2 more references, one is an example of het-erogeneities in plumes, and the other is an example of successful use of drones during high risk scenarios;

"Kelly, P.J., Kern, C., Roberts, T.J., Lopez, T., Werner, C., Aiuppa, A., 2013. Rapid chemical evolution of tropospheric volcanic emissions from Redoubt Volcano, Alaska, based on observations of ozone and halogen-containing gases. J. Volcanol. Geotherm. Res. 259, 317–333. https://doi.org/10.1016/J.JVOLGEORES.2012.04.023"

"Syahbana, D.K., Kasbani, K., Suantika, G., Prambada, O., Andreas, A.S., Saing, U.B., Kunrat, S.L., Andreastuti, S., Martanto, M., Kriswati, E., Suparman, Y., Humaida, H., Ogburn, S., Kelly, P.J., Wellik, J., Wright, H.M.N., Pesicek, J.D., Wessels, R., Kern, C., Lisowski, M., Diefenbach, A., Poland, M., Beauducel, F., Pallister, J., Vaughan, R.G., Lowenstern, J.B., 2019. The 2017–19 activity at Mount Agung in Bali (Indonesia): Intense unrest, monitoring, crisis response, evacuation, and eruption. Sci. Rep. 9, 8848. https://doi.org/10.1038/s41598-019-45295-9"

Please also note the supplement to this comment:
https://amt.copernicus.org/preprints/amt-2020-452/amt-2020-452-AC1-supplement.pdf
* * *
[Figure]

[Figure]

**Fig. 1.** This is a new Fig.1. showing some flight log data

[Figure]

**Fig. 2.** This is the new Fig.2., replacing the earlier Fig.1.

[Figure]

**Fig. 3.** Figure illustrating closed loop gas-bag measurements, Fig S.1. in Supplement

---

## Author Comment (AC2) · 6 Mar 2021

Response to questions and suggestions from Reviewer 2

Reviewer: Page 1, line 22: Instead of ". . .multi-copter drone. . ." it seems more appropriate to use the term ". . .multicopter drone. . ." or – as in the title of the manuscript – ". . .multi-rotor drone. . .".

Response: Done! Multi-rotor drone is consistently used.

Reviewer: Page 2, line 56: Instead of ". . .Mori (2016). . ." it should read ". . .Mori et al. (2016). . .". Instead of ". . .multi-rotor. . ." it seems more appropriate to use the

term ". . .multi-rotor drone. . ." consistently – also in several other sections.

Response: Done!

Reviewer: Page 4, line 150: At first glance, the presented finding that a balance of "rise and forward motion" is more favourable in terms of energy consumption than "moving only in one direction at a time" seems to be obvious. But possibly a rule for the optimal balance between "rise and forward motion" in terms of minimum energy consumption has been identified. If applicable, this should be specified.

Response: More details on this are given and are illustrated by a graph showing flight-logs for a typical flight (Fig.1.). "When ascending and moving horizontally, it was found that energy consumption could be reduced if the rise and forward motion was balanced in an optimal way, as compared to moving only in one direction at a time. This is because a considerable horizontal component in the movement gives a lift that reduces the energy consumption for maintaining the vertical position. An additional advantage is that the drone then fly in undisturbed air with less turbulence compared to a clean vertical movement."

Reviewer: Page 5, line 179: Instead of "longer propellers" it seems more appropriate to use the term "larger propellers" or "propellers with a larger diameter" – also page 24, line 659.

Response: Done!

Reviewer: Page 7, line 217 – drone drift method: Even if there is no side wind, a multi-rotor drone drifts slightly in one direction when GPS lock mode is deactivated. Has it been investigated how large this offset is and was this taken into account when measuring the plume speed ?

Response: We have not noticed this effect when hoovering at calm conditions and thus has not specifically studied this effect. We only use horizontal wind speed taken from the drone, as wind direction is taken from the MobileDOAS traverse intersection of the

plume, assuming a straight line between the source and the point of maximum gas column.

Reviewer: Does the drift speed determined from the GPS data also include a vertical speed com- ponent? Or does the drone only drift in a lateral direction and maintain the position in the vertical?

Response: The drone keep its altitude and we only use the horizontal component of the wind.

Reviewer: Page 7, line 222 – onboard anemometer: According to Appendix A, an FT205EV anemometer has been applied onboard the drone. Please specify whether this anemometer measures only the horizontal or also the vertical component of the wind speed ?

Response: The anemometer only measures horizontal wind.

Reviewer: Has it been investigated whether - and if so to what extent - the wind mea- surement using the anemometer mounted on top of the multi-rotor drone was influ- enced by the air flow created by the propellers ?

Response: We have not investigated this, except checking that there is no influence when hoovering at ground.

Reviewer: In addition to the photo in Fig. 1, it would be useful to have a sketch show- ing the exact location of the onboard anemometer and in particular its horizontal and vertical distances from the propellers.

Response: We have not included a specific figure for this but instead improved Fig 1 (now Fig 2) to include the location of the anemometer. This new figure is attached here as Fig.2.

Reviewer: Page 8, line 245: Please check "...described in section 2.3.1" since this section does not seem to exist.

Response: Changed to section 2.2

Reviewer: Page 11, line 307: The wording ". . .using homemade software. . ." is ambiguous, as it could be understood to mean that the software is developed by a software provider with the name "homemade" (which exists); contrary to that it might be intended to indicate that the software is "self-developed". Please clarify if necessary .

Response: Changed to "self-developed"

Reviewer: Page 11, line 316: Instead of ". . .a rapidly fluctuation signal is measured. . ." it should read ". . .a rapidly fluctuating signal is measured . . .".

Response: Changed to "fluctuating"

Reviewer: Page 11, line 325: The wording ". . .a practical solution is to take a sample of a time- varying signal and then expose the sensors to the sampled gas. . ." is possibly inappro- priate. Maybe what is meant is that "a sample of gas is taken" and then the sensors are exposed to this sample of gas. Please consider this and amend the wording corre- spondingly, if necessary.

Response: Changed as suggested

Reviewer: Page 11, line 326: It is stated that "Our system fulfills these two criteria: the sensors have similar response characteristics. . ." while on page 12, line 345 it is stated that "Because our sensors operate according to different principles, the sensor response times are usually different;. . .". Please clarify whether different sensors are meant in each case.

Response: The text referred to on page 12 have been changed to "Because sensors often operate. . ." to clarify that this is a general statement as compared to the text on page 11 that refers to our system in specific.

Reviewer: Page 12, line 359: It is stated that "Such dynamic changes (with frequency

components of higher than 0.5 Hz) in plume composition are assumed to be improbable for most typical scenarios". Are there any published studies or own measurement results on this subject ?

Response: No specific study has been made on this. This sentence is based on personal experiences only.

Reviewer: Page 15, line 444 – small rotary pump: Some of these small rotary pumps have vanes made of graphite, which can cause carbonaceous abrasion. Has it been investigated whether using such a pump influences the gas composition and isotopic analysis ?

Response: No, this has not been investigated by us. We have not recognized that this may be an issue... ' Reviewer: Page 18, line 530: Please indicate whether the plume speed measured using the anemometer when "the drone is kept in a fixed position" is the horizontal plume speed component only or the sum of horizontal and vertical plume speed components, i.e. including the buoyance of the plume .

Response: The anemometer measures horizontal wind only, as also the other methods and model. This has been clarified in text and Fig 9 (new Fig 10)

Reviewer: Page 19, line 538: Please indicate whether the plume speed measured using the drone drift method is the horizontal plume speed component only or the sum of horizontal and vertical plume speed components, i.e. including the buoyance of the plume.

Response: Only the horizontal components are evaluated in the drift method. In principle it would be possible to also measure the vertical component, but only the horizontal component is used in the emission rate measurements.

Reviewer: Page 20, line 556: Please clarify that the altitude is "1000 m AMSL". Page 20, line 561: A reference is missing in the caption of Fig. 10. Page 22, line 590: A reference is missing in the caption of Fig. 12. Page 24, line 653: Please correct ". . .of

of. . .". Page 24, line 655: Please correct ". . .the the . . .".

Response: All these remarks are recognized and corrected.

Reviewer: Page 25, line 677: The trajectories show remarkably long flight distances in both horizontal and vertical directions, especially considering the relatively small drone size. Please indicate whether the drone was manually controlled only during these flight distances and, if so, whether there was any support for the pilot, for example through onboard cameras.

Response: Yes, we agree that the long flight distances are remarkable. This, and the operability demonstrated, are some of the main justifications of this paper. All flights were manually operated and about one third of the flights were assisted by an on-board camera. The camera was mainly used to make it possible to avoid clouds and helped the pilot to keep track of the flight parameters (goggles). More on this is included on page 7. " Camera: During the later part of the campaign at Manam it was found to be useful to include a camera running in FPV (First Person View ) mode. The main reason for this was that it facilitated the avoidance of clouds and thereby reduced energy consumption. It also improved the maneauverability as it gave the pilot access to critical parameters in real time within his view (goggles)."

Reviewer: Was the multi-rotor drone also flown occasionally through a volcanic ash cloud? If so, did this have any negative impact on the measuring instruments or the drone, e.g. wear on the rotor blades of the drone ?

Response: The drone was flown through volcanic clouds. Its unclear how much ash they contained. No wear on the hardware was noticed at site. However, a couple of months after return home, severe wear on the motors due to acidity, was noticed. To play safe the motors were replaced.

Reviewer: Page 26, line 718: Has the cited and listed reference "ARELLANO et al. (2016)" already been published or is it otherwise available online ?

[Figure]

Response: The reference is a report from a field campaign in 2016, which is free to access through this link: https://research.chalmers.se/publication/254380. However, there is probably not enough detail in this report to fully implement the method. To simplify this, we have shared the essential steps of the code in the Appendix.

Please also note the supplement to this comment:
https://amt.copernicus.org/preprints/amt-2020-452/amt-2020-452-AC2-supplement.pdf
* * *
[Figure]

**Fig. 1.** New Fig.1 showing an example of drone flight log

[Figure]

Fig. 2. This is the new Fig.2., replacing the earlier Fig.1.

[Figure]

**Fig. 3.** Figure illustrating closed loop gas-bag measurements, Fig S.1. in Supplement

**Supplement:**

**A multi-purpose, multi-rotor drone system for long range and high-altitude volcanic gas plume measurements**

Bo Galle[1], Santiago Arellano[1*], Nicole Bobrowski[2,3], Vladimir Conde[1], Tobias P. Fischer[4], Gustav Gerdes[5], Alexandra Gutmann[6], Thorsten Hoffmann[6], Ima Itikarai[7], Tomas Krejci[8], Emma J. Liu[9,10], Kila Mulina[7], Scott Nowicki[4,11], Tom Richardson[12], Julian Rüdiger[6], Kieran Wood[12], Jiazhi Xu[1]

[1]Department of Earth, Space and Environment, Chalmers University of Technology, SE 41296, Gothenburg, Sweden
[2]Institute for Environmental Physics, University of Heidelberg, D-69120 Heidelberg, Germany
[3]Max-Planck Institute for Chemistry, 55128, Mainz, Germany
[4]Department of Earth and Planetary Sciences, University of New Mexico, 87131, Albuquerque, NM, United States
[5]GerdesSolutions AB, 128 41, Stockholm, Sweden
[6]Department of Chemistry, Johannes Gutenberg-University, 55099, Mainz, Germany
[7]Rabaul Volcano Observatory, P.O. Box 386, Rabaul, Papua New Guinea
[8]HAB Electronic AB, 34140, Ljungby, Sweden
[9]Department of Earth Sciences, University of Cambridge, CB2 3EQ, Cambridge, United Kingdom
[10]Department of Earth Sciences, WC1E 6BS, University College London, London, United Kingdom
[11]Quantum Spatial, Inc. Albuquerque, NM, United States
[12]Department of Aerospace Engineering, University of Bristol, BS8 1TR, Bristol, United Kingdom

*Correspondence to*: Santiago Arellano (santiago.arellano@chalmers.se)

**Supplement**

**S1. Data files**

The Excel-book file amt-2020-452-raw_data contains all data used for the figures presented in this article. Each sheet is named according to the figures in which data was used and the instrument from which data was obtained, e.g. data_fig11&14_MultiGAS for the data used for figures 11 and 14 using the MultiGAS (Chalmers) instrument.

Most sheets contain the full dataset coming from the instrument, not all parameters are needed for the data presented in the figures of the article. Data that was used for the figures are marked with an orange/pink background.

The following descriptions correspond to the columns on each data-sheet:

**data_Fig1_UAV**

Data file extracted from a "ulog" file containing a series of parameters from the multi-rotor navigation sensors. The plotted columns contain:

Time stamp in μs

Roll angle

Pitch angle

Yaw angle

Thrust factor

Latitude UTM

Longitude UTM

x-distance from reference position

y-distance from reference position

z-distance from reference position (inverse of altitude)

**data_fig5_Sunkist:**

Data file as recorded by the Sunkist instrument. It contains:

Date UTC

Time UTC

$CO_2$ data at instrument resolution (calibration from U. Mainz) in ppm

$SO_2$ data at instrument resolution (calibration from U. Mainz) in ppm

Temperature in °C

Pressure in Pa

Relative humidity in %

**data_fig8_MultiGAS, data_fig12&15_MultiGAS, and data_figS1_MultiGAS**

Data files as recorded by the MultiGAS instrument, stored in internal memory, and sent to ground-station in real-time. It contains:

Time UTC

Date UTC

Lat UTM

Lon UTM

Temperature in °C

Pressure in Pa

Relative humidity in %

Altitude in m ASL

$SO_2$ signal (uncalibrated) from working electrode in ppb

H2S signal (uncalibrated) from working electrode in ppb

NO signal (uncalibrated) from working electrode in ppb

$NO_2$ signal (uncalibrated) from working electrode in ppb

$NO_2$ signal (uncalibrated) from auxiliary electrode in ppb

H2S signal (uncalibrated) from auxiliary electrode in ppb

NO signal (uncalibrated) from auxiliary electrode in ppb

$SO_2$ signal (uncalibrated) from auxiliary electrode in ppb

$CO_2$ signal (uncalibrated) from auxiliary electrode in ppm

x-tilt angle in deg

y-tilt angle in deg

z-tilt angle in deg

Wind speed from anemometer in m/s

Wind direction from anemometer in deg

Activation of pump for bag sampling (0/1)

**data_fig9_UAV**

Data file extracted from a "ulog" file containing a series of parameters from the multi-rotor navigation sensors. The plotted columns contain:

Time stamp in µs

Horizontal wind speed, calculated from components x and y of wind speed, in m/s

**data_fig10_WindSpeed**

Time series of wind speed obtained from three different methods: (i) modeled data from ECMWF ERA5 Re-analysis database (C3S, 2017) retrieved at hourly resolution, 0.25×0.25 deg horizontal resolution, and 16 pressure-levels from ground to about 10 km ASL. This data is then interpolated at the location of the summit of Manam. (ii) Wind data measured by the drone using anemometer and drift-method. (iii) Wind data measured from ground using the dual-beam method described in Johansson et al. (2009). The columns contain:

Date UTC

Time UTC

Horizontal component of wind speed in m/s

**data_fig11&15_MobileDOAS**

Data file produced from an evaluation with the MobileDOAS software (Johansson et al., 2010) of data collected by the instrument during a traverse with the drone. It contains

Time UTC

Latitude UTM

Longitue UTM

Altitude m ASL

$SO_2$ vertical column density in ppm*m

**data_fig13_MultiGAS**

Data from MultiGAS instrument after calibration and other corrections. It contains

$SO_2$ volume mixing ratio in plume in ppm

$CO_2$ volume mixing ration in plume in ppm

**S2. Proof of concept of MultiGAS measurement with stabilized sample taken by the drone**

A sample of the plume was taken by filling a Tedlar bag through remote activation of a pump when a signal of $SO_2$ was detected and then connected in closed loop to the MultiGAS sensor for several minutes after landing. This sampling method was applied several times during the field campaign in Manam in May 2019, but all samples with concentration of $SO_2$ higher than 10 ppm were used for attempting measurements of the isotopic composition of carbon.

The results of the 'practice' measurement are shown in Figure S1.

[Figure]

**Figure S1. Time series of uncalibrated signals observed in the field on 22 May 2019 after connecting a Tedlar bag filled by remote activation of a pump onboard the drone to the MultiGAS instrument in closed-loop**

As clearly shown, the $SO_2$ signal is detectable and its curve of growth stabilized in about 30 s ($t_{90}$), which matches well with the manufacturer's specifications. But the signal, after calibration, is less than 1 ppm and therefore the $CO_2$ signal above background is below the detection limit of the instrument (remember the molar ratio for both species at Manam was found to be ~1).

The figure also shows a noise picked up by the $CO_2$ sensor, which we attribute to the radio. This signal is subtracted together with the background as part of the corrections.

**S3. Calibration tests of MultiGAS $CO_2$ and $SO_2$ sensors**

Calibration tests were performed to characterize the responses of the $CO_2$ and $SO_2$ sensors for the MultiGAS instrument (Xu, 2019). We used a mixture of $CO_2$ and $SO_2$ at nominal concentrations of $4.293 \pm 0.086\%$ (mol) and $203.9 \pm 4.1$ ppm (mol), respectively (i.e. in a ratio of 210.5:1), and mix it with pure $N_2$ to prepare diluted mixtures at 0, 0.5, 1, 2, and 4 ppm $SO_2$. The mixture was controlled by a dynamic gas calibrator (Thermo Scientific, model 146i) with a flow rate of 5 l/min. The mixture was pumped into the inlet of the MultiGAS at a constant flow rate of 0.5 l/min and ambient temperature (27.8°C) and pressure (102 kPa). Besides the MultiGAS system used in Manam (referred as "flow through" in the figures below, due to the use of an active pump), another system with the sensors exposed passively to the gas ("diffusion") was tested, and they were compared to a reference system using a more precise instrument based on a LI-COR 7200 sensor, which has a time response $t_{90}$ of 0.1 s and 0.3 ppm precision ("sniffer"). Each mixture was measured for periods of about 30 minutes until the gas calibrator was stabilized. The results of the measurements are shown in Figure S2.

[Figure]

**Figure S2. Calibration measurements for the MultiGAS instrument used in Manam (flow-through), in comparison with two another model of MultiGAS with sensors directly exposed to the gas without a pump (diffusion) and with a reference instrument based on a LI-COR 7200 sensor. The instruments were exposed to known concentrations of gas from which the calibration constants (offset and sensitivity) were derived. The upper panels show the measured signals using the calibration constants from the manufacturer, the middle panels show the calibration curves to derive the effective constants and the lower panels showed the calibrated signals. The flow-through instrument showed much lower noise and it was chosen for the measurements at Manam (figures from Xu, 2019).**

Additionally, an experiment was designed to characterize the time response of the sensors. For this, the instruments were placed inside a box with a stable mixture of $CO_2$ and $SO_2$ and then the instruments were suddenly removed out of the box to be exposed to ambient concentrations of these gases. This sudden change in concentration mimics a step calibration function. From the decay in signal it was possible to estimate the response times at 90% level ($t_{90}$) of the sensors. The experiment was repeated two times and responses times of 20-25 s and 20 s were found for the MultiGAS (flow-through) $SO_2$ and $CO_2$ sensors, respectively. The results are shown in Figure S3.

[Figure]

[Figure]

(a) First time.  (b) Second time.

**Figure S3. Response-time experiments showing the decay in signal when the sensors of the MultiGAS instruments (diffusion and flow-through) were exposed to a stable concentration of SO2 and SO2 and then removed suddenly to ambient concentration. Notice the quicker response of the system using a pump (flow-through, used in Manam) and the stabilization after response times of 20-25 s for $SO_2$ and 20 s for $CO_2$.**

**References**

Copernicus Climate Change Service (C3S) (2017): ERA5: Fifth generation of ECMWF atmospheric reanalyses of the global climate. Copernicus Climate Change Service Climate Data Store (CDS), *accessed in May 2019*. https://cds.climate.copernicus.eu/cdsapp#!/home

Johansson, M., Galle, B., Zhang, Y. *et al.* The dual-beam mini-DOAS technique—measurements of volcanic gas emission, plume height and plume speed with a single instrument. *Bull Volcanol* **71,** 747–751 (2009). https://doi.org/10.1007/s00445-008-0260-8

Johansson, M., MobileDOAS software, Chalmers University of Technology, 2010.

Xu, J., Development of A Drone Concept for Volcanic Gas Measurements, MSc. Thesis, Chalmers University of Technology, 2019.